# Semantic Visual Navigation by Watching YouTube Videos

**Matthew Chang**    **Arjun Gupta**    **Saurabh Gupta**
University of Illinois at Urbana-Champaign
{mc48, arjung2, saurabhg}@illinois.edu

## Abstract

Semantic cues and statistical regularities in real-world environment layouts can improve efficiency for navigation in novel environments. This paper learns and leverages such semantic cues for navigating to objects of interest in novel environments, by simply watching YouTube videos. This is challenging because YouTube videos don't come with labels for actions or goals, and may not even showcase optimal behavior. Our method tackles these challenges through the use of Q-learning on pseudo-labeled transition quadruples (image, action, next image, reward). We show that such off-policy Q-learning from passive data is able to learn meaningful semantic cues for navigation. These cues, when used in a hierarchical navigation policy, lead to improved efficiency at the ObjectGoal task in visually realistic simulations. We observe a relative improvement of $15 - 83\%$ over end-to-end RL, behavior cloning, and classical methods, while using minimal direct interaction.

## 1 Introduction

Consider the task of finding your way to the bathroom while at a new restaurant. As humans, we can efficiently solve such tasks in novel environments in a zero-shot manner. We leverage common sense patterns in the layout of environments, which we have built from our past experience of similar environments. For finding a bathroom, such cues will be that they are typically towards the back of the restaurant, away from the main seating area, behind a corner, and might have signs pointing to their locations (see Figure 1). Building computational systems that can similarly leverage such semantic regularities for navigation has been a long-standing goal.

Hand-specifying what these semantic cues are, and how they should be used by a navigation policy is challenging. Thus, the dominant paradigm is to directly learn what these cues are, and how to use them for navigation tasks, in an end-to-end manner via reinforcement learning. While this is a promising approach to this problem, it is sample inefficient, and requires many million interaction samples with dense reward signals to learn reasonable policies.

But, is this the most direct and efficient way of learning about such semantic cues? At the end of the day, these semantic cues are just based upon spatial consistency in co-occurrence of visual patterns next to one another. That is, if there is always a bathroom around the corner towards the back of the restaurant, then we can learn to find this bathroom, by simply finding corners towards the back of the restaurant. This observation motivates our work, where we pursue an alternate paradigm to learn semantic cues for navigation: learning about this spatial co-occurrence in indoor environments through video tours of indoor spaces. People upload such videos to YouTube (see project video) to showcase real estate for renting and selling. We develop techniques that leverage such YouTube videos to learn semantic cues for effective navigation to semantic targets in indoor home environments (such as finding a *bed* or a *toilet*).

---

Project website with code, models, and videos: https://matthewchang.github.io/value-learning-from-videos/.

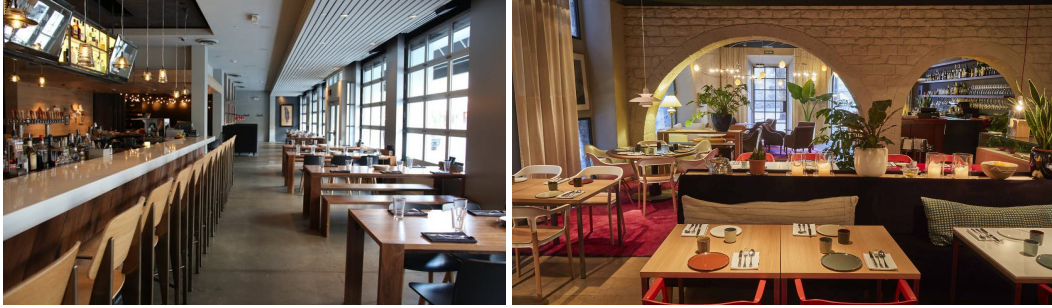

**Figure 1: Semantic Cues for Navigation.** Even though you don't see a restroom, or a sign pointing to one in either of these images, going straight ahead in the left image is more likely to lead to a restroom than going straight in the right image. This paper seeks to learn and levarage such semantic cues for finding objects in novel environments, by watching egocentric YouTube videos.

Such use of videos presents three unique and novel challenges, that don't arise in standard learning from demonstration. Unlike robotic demonstrations, videos on the Internet don't come with any action labels. This precludes learning from demonstration or imitation learning. Furthermore, goals and intents depicted in videos are not known, *i.e.*, we don't apriori know what each trajectory is a demonstration for. Even if we were to label this somehow, the depicted trajectories may not be optimal, a critical assumption in learning from demonstration [53] or inverse reinforcement learning [43].

Our formulation, *Value Learning from Videos* or *VLV*, tackles these problems by **a)** using pseudo action labels obtained by running an inverse model, and **b)** employing Q-learning to learn from video sequences that have been pseudo-labeled with actions. We follow work from Kumar *et al.* [38] and use a small number of interaction samples ($40K$) to acquire an inverse model. This inverse model is used to pseudo-label consecutive video frames with the action the robot would have taken to induce a similar view change. This tackles the problem of missing actions. Next, we obtain goal labels by classifying video frames based on whether or not they contain the desired target objects. Such labeling can be done using off-the shelf object detectors. Use of Q-learning [64] with consecutive frames, intervening actions (from inverse model), and rewards (from object category labels), leads to learning *optimal* Q-functions for reaching goals [59,64]. We take the maximum Q-value over all actions, to obtain value functions. These value functions are exactly $\gamma^s$, where $s$ is the number of steps to the nearest view location of the object of interest ($\gamma$ is the Q-learning discount factor). These value functions implicitly learn semantic cues. An image looking at the corner towards the back of the restaurant will have a higher value (for *bathroom* as the semantic target) than an image looking at the entrance of the restaurant. These learned value functions when used with a hierarchical navigation policy, efficiently guide locomotion controllers to desired semantic targets in the environment.

Learning from such videos can have many advantages, some of which address limitations of learning from direct interaction (such as via RL). Learning from direct interaction suffers from high sample complexity (the policy needs to discover high-reward trajectories which may be hard to find in sparse reward scenarios) and poor generalization (limited number of instrumented physical environments available for reward-based learning, or sim2real gap). Learning from videos side-steps both these issues. We observe a $47 - 83\%$ relative improvement in performance over RL and imitation learning methods, while also improving upon strong classical methods.

## 2   Related Work

This paper tackles semantic visual navigation in novel environments. Our proposed solution is a hierarchical policy that employs value functions learned from videos. We survey different navigation tasks, the different representations used to tackle them, and the different training methodologies employed to build those representations.

**Navigation Tasks.** Navigation tasks take many forms [3], but can largely be grouped into two categories based on whether they require exploration or not. Finding paths in known environments [70], or going to a known relative offset in a previously unknown environment [27], do not require very much exploration. On the other hand, tasks such as finding an object [27] (or a given image target [11]) in

a novel environment, or exhaustively mapping one [10, 12], require exploration and are thus more challenging. Our down-stream task of finding objects in previously unseen novel environments falls into this second category. Most current work [16, 27, 44, 69] on this task employ end-to-end, interaction-heavy learning to get at necessary semantic cues. Our work instead seeks to mine them from videos with minimal active interaction.

**Representations.** Solving navigation tasks, requires building and maintaining representations for space. These range from explicit metric maps [20, 60, 67] or topological representations [11, 35, 51], to more abstract learned implicit representations [41]. Such learned representations can effectively learn about semantic cues. Research has also focused on making classical metric and topological representations more semantic: explicitly by storing object detector or scene classifier outputs [8, 25, 30, 36, 42, 46, 66], or implicitly by storing abstract learned feature vectors useful for the end-task [27]. In our work, we use a hybrid topological and metric representation that incorporates implicit semantic information. Our focus is on investigating alternate ways of learning such semantic information.

**Hierarchical Policies.** Researchers have pursued many different hierarchical policies [7] for navigation: no hierarchy [41], macro-actions [27, 70], low-level controllers [6, 33], and sub-policies [10, 15, 25]. In particular, Chaplot *et al.* [10, 11] decompose exploration policies into a global policy, for high-level semantic reasoning, and a local policy, for low-level execution to achieve short-term goals produced by the global policy. We follow a similar decomposition, but tackle a different task (object goal), and investigate learning from unlabeled passive data *vs.* active interaction or strong supervision.

**Training Methodology.** Different papers pursue different strategies for training navigation policies: no training [60], supervised learning for collision avoidance [22, 24], behavior cloning, DAgger [27, 37, 48], reinforcement learning with sparse and dense rewards [10, 41, 49, 50, 65, 70], and combinations of imitation and RL [12, 14, 47]. In contrast, this paper designs a technique to derive navigation policies by watching YouTube videos. This is most similar to work from Kumar *et al.* [38] that studies how to learn low-level locomotion sub-routines from *synthetic* videos. In contrast, we learn high-level semantic cues from actual YouTube videos.

**Learning for Acting from Videos.** Learning about affordances [21], state-transitions [2, 31], and task-solving procedures [13], with the goal of aiding learning for robots, is a long-standing goal in computer vision. Our work is also a step in this direction, although our output is directly useful for building navigation policies, and our experiments demonstrate this.

**Learning without Action Labels.** A number of recent papers focus on learning from observation-only (or state-only) demonstrations (*i.e.* demonstrations without action labels). Some works focus on directly learning policies from such data [19, 23, 45, 54, 61, 62], while others focus on extracting a reward function for subsequent policy learning through RL [5, 17, 18, 40, 56, 57]. All of these works focus on learning a policy for the *same* task in the *same* environment that is depicted in the observation-only demonstrations (with the exception of Gangwani *et al.* [23] who show results in MDPs with different transition dynamics). Our work relaxes both these assumptions, and we are able to use video sequences to derive cues that aid solving novel tasks in novel environments.

## 3    Proposed Approach

The final task we tackle is that of reaching semantic goals in a novel environment, *i.e.*, at test time we will place the agent in a novel environment and measure how efficiently it can find common house-hold objects (bed, chair, sofa, table and toilets).

**Overview.** We design a 2-level hierarchical policy. The *high-level policy* incrementally builds a topological graph and uses *semantic reasoning* to pick promising directions of exploration. It generates a short-term goal (within $2m$) for the *low-level policy*, that achieves it or returns that the short-term goal is infeasible. This process is repeated till the agent reaches its goal. We describe the details of this hierarchical policy in Supplementary Section S1. Our central contribution is the procedure for learning the semantic reasoning function, which we call a *value* function (following RL terminology [59]), for the high-level policy from videos, and we describe this next.

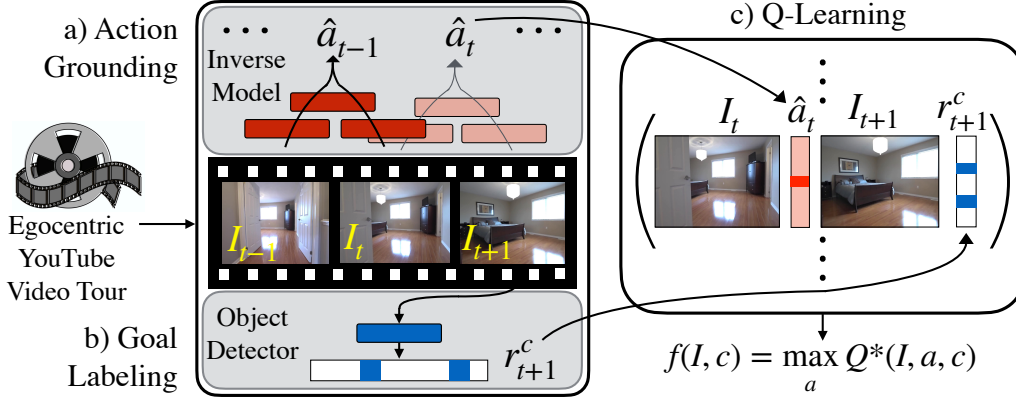

**Figure 2: Learning Values Functions from Videos.** Egocentric videos tours of indoor spaces are **a)** grounded in actions (by labeling via an inverse model), **b)** labeled with goals (using an object detector). This prepares them for **c)** Q-learning, which can extract out optimal Q-functions for reaching goals purely by watching egocentric videos. See Section 3.1 for more details.

## 3.1 Value Learning from Videos

Given an image $I$ and a set of object categories $\mathcal{C}$, we seek to learn a function $f(I, c)$ that can predict the *value* for image $I$ for reaching an object of category $c \in \mathcal{C}$. Images that show physical space close to instances of category $c$ should have a higher value than images that show regions far away from it.

Let's say we have $\mathcal{V}$, a set of egocentric video tours of indoor spaces. We seek to learn this function from such videos. We follow a three step procedure: **a)** imagining robot actions that convey the robot between intervening frames, **b)** labeling video frames of images containing instances of the desired object category, and **c)** Q-learning on the resulting reward-labeled image-action sequence trajectories. Figure 2 shows an overview of this process, we describe it in more detail below.

**Action grounding.** Such videos don't come with any information for how one image is related to another. We follow the pseudo-labeling approach from [38, 61], to imagine the actions the robotic agent would have taken to induce the depicted transformation. We collect a small amount of interaction data, where a robot executes random actions in a handful of environments. This data is in the form of image action sequences, $\ldots, I_t, a_t, I_{t+1}, \ldots$, and importantly, has information of the action that was executed to go from $I_t$ to $I_{t+1}$. We use this interaction dataset to train a *one-step inverse model* $\psi$ [1, 32] that uses $I_t$ and $I_{t+1}$ to predict $\hat{a}_t = \psi(I_t, I_{t+1})$. $\psi$ is trained via a cross-entropy loss between its prediction $\hat{a}_t$ and ground truth $a_t$. We use this inverse model $\psi$ to *pseudo-label* the video dataset $\mathcal{V}$ with action labels to obtain $\hat{\mathcal{V}}$.

**Labeling Video Frames with Goals.** Our next step involves labeling video frames with the presence of object instances from categories in $\mathcal{C}$. This can simply be done by using an off-the-shelf object detector $D$ (such as Mask RCNN [28]) trained on the MS-COCO dataset [39]. We assign a binary reward value $r^c(I)$ for each category $c$ for each video frame $I$: $+1$ if object detected, and $0$ otherwise.

**Value Learning via Off-policy Q-Learning.** Our next step is to derive value function $f(I, c)$ for the different categories. The above two steps, generate reward-labeled, image-action trajectories for traversals in indoor environments. For each category $c \in \mathcal{C}$, these are in the form of quadruples $\left(I_t, \hat{a}_t, I_{t+1}, r_{t+1}^c\right)$, where $I_t$ and $I_{t+1}$ are consecutive frames, $\hat{a}_t$ is the pseudo-label as predicted from the inverse model $\psi$, and $r_{t+1}^c$ is the label for category $c$ for image $I_{t+1}$. These quadruples can be thought of as transitions from a Markov Decision Process (MDP) [59], where the agent gets $+1$ reward for entering into a location close to the desired target object, and $0$ reward otherwise.

Thus, even though we don't have access to the physical environment, a simple video traversal of an indoor space can be pseudo-labeled to obtain transition samples from the underlying MDP operating in this environment. Under mild conditions, such samples are all that are necessary for learning *optimal* value functions via Q-learning [64]. Thus, instead of directly learning the value function $f(I, c)$, we learn a Q-function $Q(I, c, a)$ that predicts the $Q$ value of executing action $a$ when at image $I$ and seeking to find object from category $c$.

Q-learning takes the following form, where we seek to learn the fixed point, $Q^*$ of the following Bellman equation (for each category $c$): $Q^*(I_t, a_t, c) = \max_{a'} \left( r_{t+1}^c + \gamma Q^* \left( I_{t+1}, a', c \right) \right)$. This is done by finding the $Q$ that minimizes the following objective function, over transition quadruples from $\hat{\mathcal{V}}$ (we parameterize $Q$ as a convolutional neural network (more details in Section 4)):

$$\sum_{\hat{\mathcal{V}}} \left[ Q(I_t, a_t, c) - \left( r_{t+1}^c + \gamma \max_{a'} Q \left( I_{t+1}, a', c \right) \right) \right]^2. \tag{1}$$

Value function $f(I, c)$ can be obtained by simply taking a maximum of the Q-values over all actions, *i.e.*, $f(I, c) = \max_a Q(I, a, c)$. This gives us our desired value function.

Note, Q-learning can learn optimal Q-functions independent of where transition quadruples come from (as long as they cover the space), and in particular, can learn from off-policy data. This allows us to learn *optimal* value functions even though the video dataset may not follow optimal paths to any targets. This also leads us to favor Q-learning over the simpler alternative of employing Monte Carlo or TD(0) *policy evaluation* [59]. Policy evaluation is simpler as it does not involve reasoning about intervening actions, but consequently only learns the value of the underlying policy depicted in the video, rather than the optimal policy. Our experiments demonstrate this contrast between these two design choices, in scenarios where videos don't show the optimal goal reaching behavior.

The learned Q-function, and the associated value function $f(I, c)$, implicitly learn semantic cues for navigation. They can learn what images lead to the desired category, and what don't. Relative magnitude of their prediction can be used to pick directions for exploration. It is worth noting, this obtained value function is the optimal value function under the dynamics of the agent recording the video. We are implicitly assuming that optimal value function under the robot's action space or dynamics would be similar enough. This assumption may not always be true (specially at fine temporal scales), but is true in a number of situations at coarser time scales.

## 4 Experiments

We show results on the ObjectGoal task in novel environments [3]. Our experiments test the extent to which we are able to learn semantic cues for navigation by watching videos, and how this compares to alternate techniques for learning such cues via direct interaction. We also compare against alternate ways of learning from passive video data, and show visualizations of our learned value functions.

**Video Dataset.** We mined for real estate tours from YouTube. This *YouTube House Tours Dataset* consists of 1387 videos with a total run length of 119 hours. A sample video is shown in supplementary video. We sample a frame every 1.5 seconds resulting in 550K transitions tuples $I_t, I_{t+1}$ for Q-learning (after removing outdoor scenes and people). We denote this dataset as $\mathcal{V}_{yt}$.

**Experimental Setup.** We work with a simulated robot in visually realistic simulation environments. We use the Habitat simulator [52] with the Gibson environments [68] (100 training environments from the medium split, and the 5 validation environments from the tiny split). These environments are derived from scans of real world environments, and thus retain the visual and layout complexity of the real world, but at the same time allow for systematic experimentation.

We split the 105 environments into three sets: $\mathcal{E}_{\text{train}}$, $\mathcal{E}_{\text{test}}$, and $\mathcal{E}_{\text{video}}$ with 15, 5, and 85 environments respectively. The robot has access to, and can directly interact with environments in $\mathcal{E}_{\text{train}}$. $\mathcal{E}_{\text{test}}$ is same as the official Gibson tiny validation set that comes with human verified semantic class labels [4]. It is used to setup downstream semantic navigation tasks for evaluation. $\mathcal{E}_{\text{train}}$ and $\mathcal{V}_{yt}$ are used for learning via our proposed formulation. Learned policies are evaluated on $\mathcal{E}_{\text{test}}$. For some additional control experiments, we also create a dataset of synthetic videos $\mathcal{V}_{\text{syn}}$ using the 85 environments in $\mathcal{E}_{\text{video}}$ (generation procedure described in supplementary). Our splitting procedure ensures: **a)** final testing happens in novel, previously unseen environments, and **b)** the robot does not have direct access to environments in which videos were shot (neither the $\mathcal{E}_{\text{video}}$ used to generate $\mathcal{V}_{\text{syn}}$, nor the real estate shown in YouTube House Tours Dataset $\mathcal{V}_{yt}$).

**Robot Model.** We use a simplified robot action space with four actions: move forward by $25cm$, rotate left $30°$, rotate right $30°$ and stop. We assume perfect localization, that is, the robot exactly knows where it is relative to its previous location. This can be achieved by running a SLAM system,

or using additional sensors such as an IMU. The robot is a $1.25m$ long cylinder of radius $10cm$, and has a RGB-D camera with $90°$ field of view, mounted at a height of $1.25m$.

**Semantic Visual Navigation Task.** We set up the ObjectGoal task [3] in $\mathcal{E}_{\text{test}}$ for testing different models. Note that $\mathcal{E}_{\text{test}}$ is same as the Gibson tiny validation set (and does not overlap with environments in $\mathcal{E}_{\text{train}}$ or $\mathcal{E}_{\text{video}}$), and comes with human-verified annotations for semantic classes. We use these semantic annotations to set up the ObjectGoal task for 5 categories: bed, chair, couch, dining table, and toilet. We sample 1075 test episodes, equally distributed among these 5 classes. For each episode, the agent is initialized at the starting location, and asked to go to the chosen object category. An episode is considered successfully solved if the agent reaches within $1m$ of *any* instance of the target category. We report both the success rate and SPL [3]. Minimum geodesic distance to *any* instance of the target category, is used as the reference path length for computing SPL. We consider two settings: *Oracle Stop* (episode is automatically terminated and deemed successful when the agent is within $1m$ of the target category), and *Policy Stop* (agent needs to indicate that it has reached the goal). We report results along with a 90% bootstrap confidence interval.

## 4.1 Implementation Details

**Action Grounding.** Inverse model $\psi$ processes RGB images $I_t$ and $I_{t+1}$ using a ResNet-18 model [29], stacks the resulting convolutional feature maps, and further processes using 2 convolutional layers, and 2 fully connected layers to obtain the final prediction for the intervening action. We train $\psi$ on 40K interaction frames gathered by randomly executing actions in $\mathcal{E}_{\text{train}}$. This is an easy learning task, we obtain close to $96\%$ classification accuracy on a held-out validation set. We use this inverse model to pseudo-label video dataset $\mathcal{V}_{\text{yt}}$ and $\mathcal{V}_{\text{syn}}$ to obtain $\hat{\mathcal{V}}_{\text{yt}}$ and $\hat{\mathcal{V}}_{\text{syn}}$.

**Object Detectors.** We use Mask RCNN [28] trained on MS-COCO dataset [39] as our detector $D_{\text{coco}}$. Frames with detections with score in the top $10\%$ are labeled as +1 reward frames. $D_{\text{coco}}$ also predicts a foreground mask for each detection. We use it to evaluate a stopping criterion at test time.

**Q-Learning.** We represent our Q-function with ResNet 18 models, followed by 1 convolutional layer, and 2 fully connected layers with ReLU non-linearities. We use Double DQN (to prevent chronic over-estimation [63]) with Adam [34] for training the Q-networks, and set $\gamma = 0.99$. As our reward is bounded between 0 and 1, clipping target value between 0 and 1 led to more stable training.

**Semantic Navigation Policy.** High-level policy stores 12 images for each node in the topological graph (obtained by rotating 12 times by $30°$ each). It uses the learned value function, $f(I, c)$, to score these $12n$ images (for a $n$ node topological graph), and samples the most promising direction for seeking objects of category $c$. The sampled direction is converted into a short-term goal by sampling a location at an offset of $1.5m$ from the chosen node's location, in the chosen view's direction. Low-level policy [26] uses occupancy maps (built using depth images) [20] with fast marching planning [58] to execute robot actions to reach the short-term goal. It returns control on success / failure / timeout. The High-level policy also factors in the distance to the sampled direction, and score from $D_{\text{coco}}$ while sampling directions. Stopping criterion: The agent chooses to stop if $D_{\text{coco}}$ fires with confidence $\geq \tau_c$ and median depth value in the predicted mask is $\leq d_c$ distance. More details are provided in Supplementary Section S1.

## 4.2 Results

Table 1 reports performance on the ObjectGoal task for our method and compares it to other methods for solving this task. An important aspect to consider is the amount and type of supervision being used by different methods. We explicitly note the scale (number of frames, environments) and type (reward signals) of active interaction used by the different methods. For *Policy Stop* setting, for all methods, we found our stopping criterion to work much better than using the method's own stop signal. We use it for all methods. Using only 40K reward-less interaction samples from $\mathcal{E}_{\text{train}}$, along with in-the-wild YouTube videos our proposed method is able to achieve an OS-SPL (Oracle Stop SPL) of 0.53 and PS-SPL (Policy Stop SPL) of 0.22 respectively in the Oracle and Policy stop settings. We put this in context of results from other methods.

**Topological Exploration** exhaustively explores the environment. It uses our hierarchical policy but replaces $f(I, c)$ with a random function, and ignores scores from $D_{\text{coco}}$ to score different directions. As the topological map grows, this baselines systematically and exhaustively explores the environment.

**Table 1: Results**: Performance for ObjectGoal in novel environments $\mathcal{E}_{\text{test}}$. Details in Section 4.2.

| Method | Training Supervision | | | Oracle Stop | | Policy Stop (using $D_{\text{coco}}$) | |
|---|---|---|---|---|---|---|---|
| | # Active Frames | Reward | Other | SPL | Success (SR) | SPL | Success (SR) |
| Topological Exploration | - | - | - | $0.30 \pm 0.02$ | $0.67 \pm 0.02$ | $0.13 \pm 0.01$ | $0.29 \pm 0.02$ |
| Detection Seeker | - | - | - | $0.46 \pm 0.02$ | $0.75 \pm 0.02$ | $0.19 \pm 0.02$ | $0.37 \pm 0.02$ |
| RL (RGB-D ResNet+3CNN) | 100K ($\mathcal{E}_{\text{train}}$) | Sparse | - | $0.17 \pm 0.01$ | $0.37 \pm 0.02$ | | |
| RL (RGB-D ResNet+3CNN) | 10M ($\mathcal{E}_{\text{train}} \cup \mathcal{E}_{\text{video}}$) | Dense | - | $0.26 \pm 0.02$ | $0.54 \pm 0.02$ | | |
| RL (RGB-D 3CNN) | 38M ($\mathcal{E}_{\text{train}} \cup \mathcal{E}_{\text{video}}$) | Dense | - | $0.28 \pm 0.02$ | $0.57 \pm 0.03$ | | |
| RL (RGB ResNet) | 20M ($\mathcal{E}_{\text{train}}$) | Dense | - | $0.29 \pm 0.02$ | $0.56 \pm 0.03$ | $0.08 \pm 0.01$ | $0.21 \pm 0.02$ |
| RL (Depth 3CNN) | 38M ($\mathcal{E}_{\text{train}}$) | Dense | - | $0.25 \pm 0.02$ | $0.52 \pm 0.02$ | | |
| Behavior Cloning | 40K ($\mathcal{E}_{\text{train}}$) | - | $\hat{\mathcal{V}}_{\text{yt}}$ | $0.25 \pm 0.02$ | $0.53 \pm 0.03$ | $0.08 \pm 0.01$ | $0.20 \pm 0.02$ |
| Behavior Cloning + RL | 12M ($\mathcal{E}_{\text{train}}$) | Dense | $\hat{\mathcal{V}}_{\text{yt}}$ | $0.24 \pm 0.02$ | $0.58 \pm 0.02$ | | |
| Our (Value Learning from Videos) | 40K ($\mathcal{E}_{\text{train}}$) | - | $\hat{\mathcal{V}}_{\text{yt}}$ | $\mathbf{0.53} \pm 0.02$ | $\mathbf{0.79} \pm 0.02$ | $\mathbf{0.22} \pm 0.02$ | $\mathbf{0.39} \pm 0.03$ |
| Behavior Cloning | 40K ($\mathcal{E}_{\text{train}}$) | - | $\hat{\mathcal{V}}_{\text{syn}}$ | $0.36 \pm 0.02$ | $0.71 \pm 0.02$ | $0.10 \pm 0.01$ | $0.26 \pm 0.02$ |
| Behavior Cloning + RL | 12M ($\mathcal{E}_{\text{train}}$) | Dense | $\hat{\mathcal{V}}_{\text{syn}}$ | $0.24 \pm 0.02$ | $0.55 \pm 0.03$ | | |
| Our (Value Learning from Videos) | 40K ($\mathcal{E}_{\text{train}}$) | - | $\hat{\mathcal{V}}_{\text{syn}}$ | $0.48 \pm 0.02$ | $0.75 \pm 0.02$ | $0.21 \pm 0.02$ | $0.38 \pm 0.03$ |
| Strong Supervision Values | Labeled Maps ($\mathcal{E}_{\text{video}}$) | | | $0.55 \pm 0.02$ | $0.81 \pm 0.02$ | $0.24 \pm 0.02$ | $0.43 \pm 0.02$ |
| Strong Supervision + VLV (Ours) | Labeled Maps ($\mathcal{E}_{\text{video}}$) + $\hat{\mathcal{V}}_{\text{yt}}$ | | | $0.57 \pm 0.02$ | $0.82 \pm 0.02$ | $0.23 \pm 0.02$ | $0.41 \pm 0.02$ |

Thus, this is quite a bit stronger than executing random actions (OS-SPL of $0.15$). It is able to find objects often (67%), though is inefficient with OS-SPL of 0.30.

**Detection Seeker** also does topological exploration, but additionally also uses scores from $D_{\text{coco}}$ to seek the object once it has been detected. This performs quite a bit better at 0.46 SPL. This indicates that object detectors provide a non-trivial signal for object goal navigation. Even lower confidence detection scores for more distant but partially visible objects will guide the agent in the right direction. Our method captures more out of view context, and consequently does better across all settings.

**End-to-end RL.** We also compare against many variants of end-to-end RL policies trained via direct interaction. We use the PPO [55] implementation for CNN+GRU policies that are implemented in Habitat [52]. We modify them to work with ObjectGoal tasks (feeding in one-hot vector for target class, modifying rewards), and most importantly adapt them to use ImageNet initialized ResNet-18 models [29] for RGB (given no standard initialization for Depth image, it is still processed using the original 3-layer CNN in Habitat code-base). The fairest comparison is to train using sparse rewards (dense rewards will require environment instrumentation not needed for our method) in $\mathcal{E}_{\text{train}}$ for 40K interaction samples with RGB-D sensors. This unsurprisingly did not work (OS-SPL: 0.17 and OS-SR: 37%). Thus, we aided this baseline by providing it combinations of more environments ($\mathcal{E}_{\text{train}} \cup \mathcal{E}_{\text{video}}$), many times more samples, and dense rewards. Even in these more favorable settings, end-to-end RL didn't perform well. The best model had a OS-SPL of 0.29 and OS-SR of 56% (*vs.* 0.50 and 75% for our method), even when given interaction access to $6\times$ more environments, $250\times$ more interaction, and dense rewards (*vs.* no rewards). This demonstrates the power of our proposed formulation that leverages YouTube videos for learning about spatial layout of environments. Policy stop evaluation is computationally expensive so, we report the score only for the strongest model.

**Behavior Cloning (BC) on Pseudo-Labeled Videos $\hat{\mathcal{V}}$.** We pre-process the videos to find trajectories that lead to objects of interest (as determined by $D_{\text{coco}}$). We train CNN+GRU models to predict the pseudo-labeled action labels on these trajectories. As this is passive data that has already been collected, we are limited to using behavior cloning wth RGB input as opposed to richer inputs or the more sophisticated DAgger [48]. This is effectively the BCO(0) [61] algorithm. This performs fairly similarly to RL methods and with negligible sample complexity, though still lags far behind our proposed method that utilizes the exact same supervision. Perhaps this is because our proposed method uses pseudo-labeled action indirectly and is more tolerant to mismatch in action space. In contrast, behavior cloning is critically reliant on action space similarity. This is brought out when we use $\hat{\mathcal{V}}_{\text{syn}}$ instead of $\hat{\mathcal{V}}_{\text{yt}}$ where the action space is more closely matched. Behavior cloning performs much better at 0.36 OS-SPL, though our method still performs better than all the baselines even when trained on videos in $\hat{\mathcal{V}}_{\text{syn}}$.

**Behavior Cloning+RL.** We also experimented with combining behavior cloning and RL. We use the behavior cloning policies obtained above, and finetune them with RL. For the same reasons as above, this policy is limited to use of RGB inputs. When finetuning from behavior cloning policy trained on

$\hat{\mathcal{V}}_{\text{yt}}$ we found performance to remain about the same (OS-SPL 0.24). When starting off from a policy trained on $\hat{\mathcal{V}}_{\text{syn}}$, we found the performance to drop to OS-SPL of 0.24. We believe that the dense reward shaped RL may be learning a qualitatively different policy than one obtained from behavior cloning. Furthermore, use of dense rewards for RL, may limit the benefit of a good initialization.

**Strong Supervision Value Function.** While our focus is on learning purely from passive data, our semantic navigation policy can also be trained using strong supervision obtained using semantically labeled maps. We train $f(I, c)$ to predict 'ground-truth' Q-values computed using the number of steps to the nearest instance of category $c$ on the meshes from environments in $\mathcal{E}_{\text{video}}$. This model is strong at OS-SPL of 0.55. This serves as a very competitive ObjectNav policy in the regime where we allow such strong supervision. Our proposed method that uses significantly less supervision (in-the-wild videos from YouTube *vs.* environment scans) is still close to the performance of this strongly supervised method (OS-SPL 0.53). When we combine the two by training the strongly supervised objective jointly with our Q-learning based objective, performance is even stronger at OS-SPL of 0.57 (significant at a p-value of 0.025).

Thus, in conclusion, value functions learned via our approach from YouTube video tours of indoor spaces are effective and efficient for semantic navigation to objects of interest in novel environments. They compare favorably to competing reinforcement learning based methods, behavior cloning approaches, and strong exploration baselines, across all metrics.

## 4.3 Ablations

We present ablations when testing policies on $\mathcal{E}_{\text{train}}$ in Oracle Stop setting. Note $\mathcal{E}_{\text{train}}$ was only used to train the inverse model, and not the Q-learning models that we seek to compare. The *base setting* from which we ablate corresponds to training $f(I, c)$ on $\hat{\mathcal{V}}_{\text{syn}}$ with pseudo-labeled actions, $D_{\text{coco}}$ based reward labels, and the use of $f(I, c)$ and spatial consistency for sampling short-term goals. This achieves an OS-SPL of $0.40 \pm 0.02$. We summarize results below, table in supplementary.

We notice only a minor impact in performance when **a)** using true actions as opposed to actions from inverse model $\psi$ ($0.41 \pm 0.03$), **b)** using true detections as opposed to detections from $D_{\text{coco}}$ ($0.40 \pm 0.03$), **c)** using true reward locations as opposed to frames from which object is visible as per $D_{\text{coco}}$ ($0.41 \pm 0.03$) (the proposed scheme treats frames with high-scoring detections as reward frames as opposed to true object locations), and **d)** using optimal trajectories as opposed to noisy trajectories ($0.43 \pm 0.03$). Albeit on simulated data, this analysis suggests that there is only a minor degradation in performance when using inferred estimates in place of ground truth values.

Perhaps, a more interesting observation is that there is a solid improvement when we additionally use $D_{\text{coco}}$ score to sample short-term goals ($0.46 \pm 0.03$). We believe use of $D_{\text{coco}}$ produces a more peak-y directional signal when the object is in direct sight, where as differences in $f(I, c)$ are more useful at long-range. Secondly, we found that use of $360°$ images at training time also leads to a strong improvement ($0.47 \pm 0.02$). We believe use of $360°$ images at training time prevents *perceptual aliasing* during Q-learning. In the base setting, Q-values can erroneously propagate via an image that looks directly at a wall. Presence of $360°$ context prevents this. While this is useful for future research, we stick with the base setting as we are limited by what videos we could find on YouTube.

**Is action pseudo-labeling necessary?** As discussed in Section 3.1, we favored use of Q-learning over action agnostic methods, such as policy evaluation, as this allows us to learn optimal value functions as opposed to value of the policy depicted in the video. To test this, we train different methods in the *branching environment* as shown in the figure on the right (top). Desired goal locations are labeled by $G_{\text{near}}$ and $G_{\text{far}}$. We investigate the learned behavior at the branch point $B$, by initializing the agent at random locations in the circle $S$. Desired behavior is for agent to reach $G_{\text{near}}$. In departure from all other experiments, here we train and test in the same branching environment. This is a deliberate choice as we seek to understand how different methods interpret the training data.

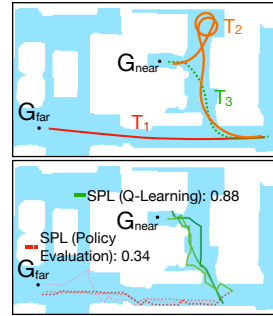

Videos in this branching environment are a $50 - 49.5 - 0.5\%$ mix of trajectories $T_1$, $T_2$, and $T_3$. $T_1$ and $T_2$ are sub-optimal trajectories to reach $G_{\text{near}}$ and $G_{\text{far}}$ respectively, while $T_3$ is the optimal trajectory to reach $G_{\text{near}}$. The policy evaluation method doesn't use any action labels, and correctly infers the values for the policy from which videos are sampled. As expected, this causes it to pursue

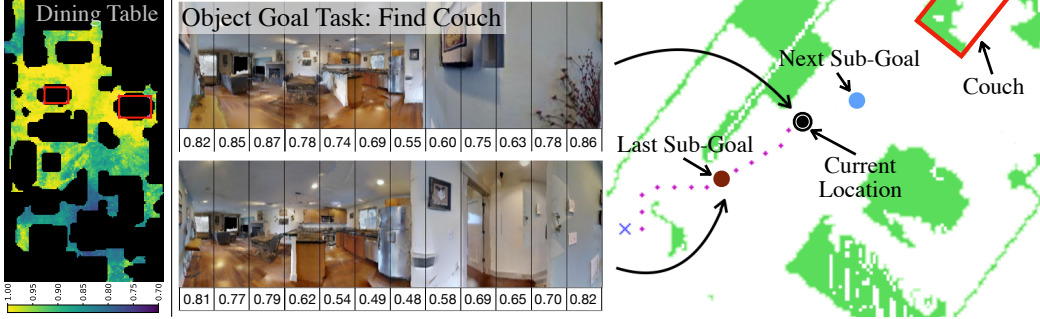

**Figure 3:** Left figure shows predicted values for reaching a *dining table* at different points on the top-view map in a novel environment. Values are high near the dining tables (denoted by the red boxes), and smoothly bleed out to farther away regions. Right shows a sample execution of our navigation policy finding a *couch* in a novel environment. More in Supplementary.

the sub-optimal goal (red paths in bottom figure). In contrast, Q-learning with pseudo-labeled actions, estimates the *optimal* value function, and consistently reaches $G_{\text{near}}$ (green paths).

## 5 Discussion

We presented a technique to enable learning of semantic cues for finding objects in novel environments from in-the-wild YouTube videos. Our proposed technique employs Q-learning on pseudo-labeled transition quadruples. This allows learning of effective semantic cues even in the absence of action grounding and goal-directed optimal behavior. When coupled with a hierarchical navigation policy, these cues convey the agent to desired objects more effectively than competitive exploration baselines and RL methods at a fraction of interaction cost. In the future, we will test our policies on real robots and extend to other navigation tasks.

## Broader Impact

Our specific research in this paper lowers barriers for the training of navigation policies. Instead of needing fully instrumented environments, or large-scale 3D scans, we can now train using video tours of indoor spaces. This significantly expands the environments that such methods can be trained on. Existing datasets [9, 68] used for training current systems have a bias towards expensive houses. This is because sensors and services involved in constructing such scans are expensive. While our current YouTube Walks dataset also has some of this bias, a video tour can be collected merely by using a phone with a camera. This will allow training of navigation policies that will work well in more typical environments, and will democratize the use of learning-based policies for navigation. We also acknowledge that the use of publicly available data from the Internet (in our case YouTube videos) raises questions about privacy and consent. These issues require a broader discussion.

Our broader research aims to improve policies for navigation in unstructured environments. This by itself has numerous desirable applications (such as automated delivery, search and monitoring in hazardous environments, automated crop inspection and mechanical weeding via under-canopy robots). Such applications can save lives, prevent food shortage (by preventing herbicide resistance), and enable development of other automation technologies.

While there are a number of critical applications that our research can potentially enable, we acknowledge that our research falls under automation, and as with all other research in this area, in the future it could replace jobs currently performed by humans. However, this must be viewed in context of the critical applications described above. Resolving or even fully understanding this trade-off will need a much broader discussion.

**Acknowledgement:** We thank Sanjeev Venkatesan for help with data collection. We also thank Rishabh Goyal, Ashish Kumar, and Tanmay Gupta for feedback on the paper. This material is based upon work supported by NSF under Grant No. IIS-2007035, and DARPA Machine Common Sense.

**Gibson dataset license:** http://svl.stanford.edu/gibson2/assets/GDS_agreement.pdf

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
