[Supplementary Material]

# Semantic Visual Navigation by Watching YouTube Videos - Supplementary Materials

**Matthew Chang**      **Arjun Gupta**      **Saurabh Gupta**
University of Illinois at Urbana-Champaign
`{mc48, arjung2, saurabhg}`@illinois.edu

## Contents

## S1   Hierarchical Policies for Semantic Navigation

We use the learned value function $f(I, c)$ from Section 3.1 (main paper) in a hierarchical navigation policy for semantic navigation. Our hierarchical policy is motivated by Chaplot *et al.* [2], and consists of a *high-level* policy and a *low-level* policy. The high-level policy outputs short-term goals that are achieved by the low-level policy. The high-level policy uses value predictions on images seen so far (at short-term goal locations), to sample a short-term goal in the most promising direction. This short-term goal is expressed as a relative offset from the agent's current location. The low-level policy

**Figure S1: Hierarchical navigation policy.** High-level policy does semantic reasoning (using the learned value functions) over images in different directions and outputs short-term goals, that are consumed by the low-level policy. The low-level policy employs classical mapping and planning to achieve the short-term goal, and returns control to the high-level policy if it achieves the short-term goal, or determines it to be infeasible. Black nodes depict nodes stored by the high-level policy in the topological graph, and blue nodes show the value predictions in different directions from each of the black nodes (size indicates predicted value, we use 12 uniformly sampled directions but only show few for clarity). Current location is indicated by the hollow circle. High-level policy outputs the most promising direction to pursue as the short-term goal. Relative offset of this location from the current location (ΔPose) is passed to the low-level policy. Low-level policy incrementally builds occupany map. It uses the fast-marching method to plan a path to the desired short-term goal, and outputs low-level robot actions. Low-level policy returns control on success (reaching the short-term goal), infeasible goal (short-term goal determined to be in occupied space), or timeout.

emits low-level robot actions to navigate to this short-term goal, or returns that the short-term goal is infeasible. This process is repeated, *i.e.*, the high-level policy takes feedback from the low-level policy, along with the image at the agent's new location to sample the next short-term goal. We describe these two policies in more detail below. Figure S1 shows an overview of this navigation policy.

## S1.1 High-level policy

The high-level policy, $\Pi$ builds a hybrid spatial and topological representation. It stores $360°$ images along with their locations at each short-term goal location. $360°$ images are obtained by incrementally rotating the agent 12 times by $30°$ each. High-level policy also stores the value prediction from $f(I, c)$ on these 12 images, for the category of interest $c$. These 12 values denote the promise of exploring in the different directions for reaching the objects of the desired class. These predicted values are combined with object detector output and a spatial consistency term to give the final score:

$$f_{comb}(I, c) = \lambda_1 f(I, c) + \lambda_2 \underbrace{\mathbb{1}_{\geq 0.5}\left[D_{coco}(I, c)\right] \cdot (1 + D_{coco}(I, c))}_{\text{Object Detector}} + 0.05\lambda_3 \underbrace{\max\left(10 - d, 0\right)}_{\text{Spatial Consistency}} \quad (1)$$

where $f(I, c)$ is the semantic score for the object class of interest $c$ on the image $I$, $D_{coco}(I, c)$ is the maximum confidence for Mask-RCNN detections of class $c$ in $I$, $d$ is the *estimated* geodesic distance (based on the current map) of the proposed short-term goal from the current agent position in meters, and $\mathbb{1}_{\geq 0.5}$ is an indicator function that outputs 1 if $D_{coco}(I, c) \geq 0.5$, and 0 otherwise. We set $\lambda_1 = \lambda_2 = \lambda_3 = 1$.

As it is expensive to get these images (it costs 12 steps), we only store these at locations where the short-term policy returns control to the high-level policy (we call these locations as *semantic reasoning locations*, and these are marked in Figure S1 with black dots).

The high-level policy maintains a priority heap of all of these $12N$ values (along with their location and associated direction vectors in the agent's coordinate frame), where $N$ is the number of semantic reasoning nodes currently stored in the topological graph. At each time step, the high-level policy pops the highest of these $12N$ values[1] from the priority heap, and samples $k \ (= 100)$ short-term

goals in this direction ($\pm 7°$) that are between $1m$ and $2m$ from the parent node. These $k$ short-term goals are passed onto the low-level policy, which pursues the first of these $k$ goals that is not known to be infeasible, and returns control to the high-level policy if it succeeds, or determines that the sampled short-term goal is infeasible or too far away.

## S1.2 Low-level policy

The low-level policy uses metric occupancy maps [3] along with fast-marching method (FMM) path planners [7] to incrementally plan paths to provided short-term goals. The low-level policy filters the provided $k$ goals for feasibility (using the current occupancy map). It takes the first one of these filtered short-term goals, plans a path to it, and outputs planned robot actions. Low-level policy continues to re-plan when the occupancy map updates. Low-level policy executes actions output from the FMM planner. It stops and returns control when **a)** it has reached the goal, **b)** it has already executed enough steps (based on estimate from original FMM computation), or **c)** the short-term goal turns out to be infeasible or much further than originally anticipated (as more of the map becomes visible). We assume access to depth images, and adapt code from the map and plan implementation from [4], to implement the low-level policy.

As our focus is on high-level semantic cues, for simplicity we assume access to perfect agent pose for this hierarchical policy. This can be achieved using additional sensors on the robot (depth cameras, and IMU units), or using a SLAM system [5], or just with RGB images by using learned pose estimators and free space estimators [1].

## S1.3 Stopping Criteria

We elaborate on the stopping criteria used for Policy Stop setting. At every semantic reasoning step, we compute a proxy measure for whether we are close to an object of the desired category or not by using the depth image and $D_{\text{coco}}$. For all high-scoring detections for class $c$ from $D_{\text{coco}}$ (detection score more than $\tau_c = 0.75$), we approximate the distance to the detected object instance by the median depth value within the predicted instance segmentation mask. If any detected instance is within a distance $d_c$, the agent emits a stop signal. $d_c$ is a per-category hyper-parameter (as object sizes vary drastically across categories). We set it using 100 episodes sampled in $\mathcal{E}_{\text{train}}$.

As noted in Section 4.2, we found that this hand-crafted stopping criteria also led to best performance for all methods that we compare to (as opposed to using the method's own stopping method). Threshold $\tau_c$ was fixed to 0.75 for all methods, while $d_c$ was optimized for each category *for each method* on the same 100 episodes from $\mathcal{E}_{\text{train}}$ using the exact same procedure. For behavior cloning and RL methods, stopping criteria is evaluated at *all* times steps, where as for our method and baselines based on our method, it is evaluated at every semantic reasoning step.

## S2 Experimental Details

### S2.1 Environment Splits

**Table S1:** List of Gibson environments in different splits. See Section 4 for details.

| Split | Environments |
|---|---|
| $\mathcal{E}_{\text{train}}$ | Andover, Annona, Adairsville, Brown, Castor, Eagan, Goodfield, Goodwine, Kemblesville, Maugansville, Nuevo, Springerville, Stilwell, Sussex |
| $\mathcal{E}_{\text{test}}$ | Collierville, Corozal, Darden, Markleeville, Wiconisco |
| $\mathcal{E}_{\text{video}}$ | Airport, Albertville, Allensville, Anaheim, Ancor, Arkansaw, Athens, Bautista, Beechwood, Benevolence, Bohemia, Bonesteel, Bonnie, Broseley, Browntown, Byers, Chilhowie, Churchton, Clairton, Coffeen, Cosmos, Cottonport, Duarte, Emmaus, Forkland, Frankfort, Globe, Goffs, Goodyear, Hainesburg, Hanson, Highspire, Hildebran, Hillsdale, Hiteman, Hominy, Irvine, Klickitat, Lakeville, Leonardo, Lindenwood, Lynchburg, Maida, Marland, Marstons, Martinville, Merom, Micanopy, Mifflinburg, Musicks, Neibert, Neshkoro, Newcomb, Newfields, Onaga, Oyens, Pamelia, Parole, Pinesdale, Pomaria, Potterville, Ranchester, Readsboro, Rogue, Rosser, Shelbiana, Shelbyville, Silas, Soldier, Stockman, Sugarville, Sunshine, Sweatman, Thrall, Tilghmanton, Timberon, Tokeland, Tolstoy, Tyler, Victorville, Wainscott, Willow, Wilseyville, Winooski, Woodbine |

### S2.2 Difficulty Distribution of Test Episodes

We plot the distribution of difficulty (distance to nearest object of interest) of the evaluation episodes in $\mathcal{E}_{\text{test}}$ in figure on right. We group these episodes into 3 difficulty levels, based on distance to the nearest instance of the target category: *easy* ($\leq 3m$, green), *medium* ($3m$ to $5m$, orange), and *hard* ($5m$ to $15m$, red). In total there were 313 easy, 324 medium and 438 hard episodes. There were 200, 250, 200, 125, 300 episodes each for object categories *Bed*, *Chair*, *Couch*, *Dining Table*, *Toilet* respectively.

Distribution of evaluation episode difficulty

### S2.3 Generation of $\mathcal{V}_{\text{syn}}$

We use environments in $\mathcal{E}_{\text{video}}$ to render out egocentric navigation tours. We employ a path planner to compute shortest path between *random* pairs of points in each environment. We render out panorama images (4 images: straight facing, left facing, back facing, and right facing, relative to the direction of motion) along these shortest paths and throw out the sequence of actions that were executed, to arrive at the dataset of videos $\mathcal{V}_{\text{syn}}$. To make these tours more realistic, we execute a random action with $20\%$ probability at each time step (and replan accordingly). We sample 300 trajectories in each of the 85 environments. Average trajectory length is 40 steps.

### S2.4 More Implementation Details

We note further implementation details for our method and baselines.

1. Topological Exploration and Detection Seeker are implemented by setting $(\lambda_1, \lambda_2, \lambda_3)$ to be $(0, 0, 1)$, and $(0, 1, 1)$ respectively in Eq. 1. This assures a fair comparison between the three methods, and tests the effectiveness of our learned function $f(I, c)$.

2. For End-to-End RL, we experimented with different architectures as noted in Table 1 in the main paper. Baselines as part of Habitat [6] use a 3 layered CNN (denoted by 3CNN and SimpleCNN interchangeably in the main paper) to represent RGB, Depth or RGB-D input. We report performance with this default network (RL (RGB-D 3CNN, RL Depth 3CNN)) in Table 1 in main paper. We found that using a ResNet-18 model (initialized by pre-training on ImageNet) worked better than using this SimpleCNN to represent RGB images. Thus we

additionally also reported performance with ResNet-18 models (RGB-D ResNet-18+3CNN, RGB ResNet-18). For RGB-D models, we could only use ResNet-18 for the RGB part. Depth is still processed through the same 3-layer CNN (as there is no standard initialization for Depth models that is commonly used). Output from ResNet-18 for RGB and 3CNN for Depth were concatenated before feeding into the LSTM model.

3. Our Q-learning models were optimized using Adam with a learning rate of $10^{-4}$, $\beta_1 = 0.9$ and $\beta_2 = 0.999$. Model was trained for $300K$ mini-batches of size 16 and the model after the last update was used for experiments.

4. **Architecture of Q-network:** The architecture of the Q-network was based off of ResNet-18. We used a ResNet-18 pretrained on ImageNet removing the last convolution layer and all later layers. We add to the pre-trained head, an additional convolution layer with kernel size $3 \times 3$ and 64 channels. After this convolution layer there are 3 fully-connected layers of size $[512, 256, 15]$ respectively. The output of the final layer is reshaped to $3 \times 5$ to represent the value of taking each of the 3 possible actions with respect to the 5 possible classes.

5. **Compute Infrastructure:** All experiments were conducted on a single GPU server with 8 GPUs (NVidia 2080 Ti). Model training for our method was done on a single GPU and took 22 hours.

## S3   Detailed Results

**Figure S2:** Oracle Stop SPL for various methods against the number of direct interaction samples used.

### S3.1   Main Results (corresponding to Section 4.1)

**Table S2: Results**: SPL and Success Rate for ObjectGoal wth **Oracle Stop** in novel environments $\mathcal{E}_{\text{test}}$ by episode difficulty. Details in Section 4.2.

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

**Table S3: Results**: SPL and Success Rate for ObjectGoal wth **Oracle Stop** in novel environments $\mathcal{E}_{\text{test}}$ by object class. Details in Section 4.2.

| Method | Training Supervision | | | SPL | | | | |
|---|---|---|---|---|---|---|---|---|
| | **# Active Frames** | **Reward** | **Other** | **Bed** | **Chair** | **Couch** | **Dining Table** | **Toilet** |
| Topological Exploration | - | - | - | $0.35 \pm 0.04$ | $0.39 \pm 0.03$ | $0.29 \pm 0.03$ | $0.27 \pm 0.04$ | $0.19 \pm 0.03$ |
| Detection Seeker | - | - | - | $\mathbf{0.49} \pm 0.05$ | $0.64 \pm 0.04$ | $0.48 \pm 0.04$ | $0.53 \pm 0.06$ | $0.26 \pm 0.03$ |
| RL (RGB-D ResNet+3CNN) | 100K ($\mathcal{E}_{\text{train}}$) | Sparse | - | $0.12 \pm 0.03$ | $0.29 \pm 0.04$ | $0.16 \pm 0.03$ | $0.20 \pm 0.05$ | $0.11 \pm 0.03$ |
| RL (RGB-D ResNet+3CNN) | 10M ($\mathcal{E}_{\text{train}} \cup \mathcal{E}_{\text{video}}$) | Dense | - | $0.24 \pm 0.04$ | $0.37 \pm 0.04$ | $0.29 \pm 0.04$ | $0.38 \pm 0.05$ | $0.09 \pm 0.02$ |
| RL (RGB-D 3CNN) | 38M ($\mathcal{E}_{\text{train}} \cup \mathcal{E}_{\text{video}}$) | Dense | - | $0.30 \pm 0.04$ | $0.39 \pm 0.04$ | $0.26 \pm 0.04$ | $0.36 \pm 0.05$ | $0.14 \pm 0.03$ |
| RL (RGB ResNet) | 20M ($\mathcal{E}_{\text{train}}$) | Dense | - | $0.30 \pm 0.04$ | $0.44 \pm 0.04$ | $0.24 \pm 0.04$ | $0.27 \pm 0.05$ | $0.21 \pm 0.03$ |
| RL (Depth 3CNN) | 38M ($\mathcal{E}_{\text{train}}$) | Dense | - | $0.29 \pm 0.04$ | $0.32 \pm 0.04$ | $0.26 \pm 0.04$ | $0.32 \pm 0.05$ | $0.13 \pm 0.02$ |
| Behavior Cloning | 40K ($\mathcal{E}_{\text{train}}$) | - | $\hat{\mathcal{V}}_{\text{yt}}$ | $0.24 \pm 0.04$ | $0.34 \pm 0.04$ | $0.28 \pm 0.04$ | $0.36 \pm 0.05$ | $0.10 \pm 0.02$ |
| Behavior Cloning + RL | 12M ($\mathcal{E}_{\text{train}}$) | Dense | $\hat{\mathcal{V}}_{\text{yt}}$ | $0.23 \pm 0.04$ | $0.33 \pm 0.03$ | $0.25 \pm 0.03$ | $0.28 \pm 0.04$ | $0.13 \pm 0.02$ |
| Our (Value Learning from Videos) | 40K ($\mathcal{E}_{\text{train}}$) | - | $\hat{\mathcal{V}}_{\text{yt}}$ | $\mathbf{0.49} \pm 0.04$ | $\mathbf{0.68} \pm 0.03$ | $\mathbf{0.60} \pm 0.04$ | $\mathbf{0.71} \pm 0.05$ | $0.32 \pm 0.03$ |
| Behavior Cloning | 40K ($\mathcal{E}_{\text{train}}$) | - | $\hat{\mathcal{V}}_{\text{syn}}$ | $0.36 \pm 0.05$ | $0.45 \pm 0.04$ | $0.37 \pm 0.04$ | $0.35 \pm 0.05$ | $0.20 \pm 0.03$ |
| Behavior Cloning + RL | 12M ($\mathcal{E}_{\text{train}}$) | Dense | $\hat{\mathcal{V}}_{\text{syn}}$ | $0.21 \pm 0.03$ | $0.31 \pm 0.03$ | $0.23 \pm 0.03$ | $0.35 \pm 0.05$ | $0.14 \pm 0.03$ |
| Our (Value Learning from Videos) | 40K ($\mathcal{E}_{\text{train}}$) | - | $\hat{\mathcal{V}}_{\text{syn}}$ | $0.44 \pm 0.04$ | $0.60 \pm 0.04$ | $0.48 \pm 0.04$ | $0.56 \pm 0.06$ | $\mathbf{0.37} \pm 0.04$ |
| Strong Supervision Values | Labeled Maps ($\mathcal{E}_{\text{video}}$) | | | $0.46 \pm 0.04$ | $0.59 \pm 0.03$ | $0.57 \pm 0.04$ | $0.68 \pm 0.05$ | $0.43 \pm 0.03$ |
| Strong Supervision + VLV (Ours) | Labeled Maps ($\mathcal{E}_{\text{video}}$) + $\hat{\mathcal{V}}_{\text{yt}}$ | | | $0.50 \pm 0.04$ | $0.69 \pm 0.03$ | $0.58 \pm 0.04$ | $0.77 \pm 0.04$ | $0.43 \pm 0.04$ |

| Method | Training Supervision | | | Success rate | | | | |
|---|---|---|---|---|---|---|---|---|
| | **# Active Frames** | **Reward** | **Other** | **Bed** | **Chair** | **Couch** | **Dining Table** | **Toilet** |
| Topological Exploration | - | - | - | $0.67 \pm 0.05$ | $0.85 \pm 0.04$ | $0.71 \pm 0.05$ | $0.68 \pm 0.07$ | $0.48 \pm 0.05$ |
| Detection Seeker | - | - | - | $0.74 \pm 0.04$ | $0.92 \pm 0.03$ | $0.80 \pm 0.05$ | $0.80 \pm 0.05$ | $0.57 \pm 0.05$ |
| RL (RGB-D ResNet+3CNN) | 100K ($\mathcal{E}_{\text{train}}$) | Sparse | - | $0.36 \pm 0.06$ | $0.54 \pm 0.05$ | $0.34 \pm 0.05$ | $0.43 \pm 0.07$ | $0.21 \pm 0.04$ |
| RL (RGB-D ResNet+3CNN) | 10M ($\mathcal{E}_{\text{train}} \cup \mathcal{E}_{\text{video}}$) | Dense | - | $0.48 \pm 0.06$ | $0.77 \pm 0.04$ | $0.62 \pm 0.05$ | $0.86 \pm 0.05$ | $0.19 \pm 0.04$ |
| RL (RGB-D 3CNN) | 38M ($\mathcal{E}_{\text{train}} \cup \mathcal{E}_{\text{video}}$) | Dense | - | $0.66 \pm 0.05$ | $0.74 \pm 0.05$ | $0.57 \pm 0.06$ | $0.74 \pm 0.07$ | $0.30 \pm 0.04$ |
| RL (RGB ResNet) | 20M ($\mathcal{E}_{\text{train}}$) | Dense | - | $0.64 \pm 0.05$ | $0.74 \pm 0.05$ | $0.55 \pm 0.06$ | $0.50 \pm 0.07$ | $0.38 \pm 0.04$ |
| RL (Depth 3CNN) | 38M ($\mathcal{E}_{\text{train}}$) | Dense | - | $0.54 \pm 0.06$ | $0.71 \pm 0.05$ | $0.54 \pm 0.06$ | $0.63 \pm 0.07$ | $0.30 \pm 0.04$ |
| Behavior Cloning | 40K ($\mathcal{E}_{\text{train}}$) | - | $\hat{\mathcal{V}}_{\text{yt}}$ | $0.46 \pm 0.06$ | $0.74 \pm 0.05$ | $0.63 \pm 0.06$ | $0.72 \pm 0.07$ | $0.24 \pm 0.04$ |
| Behavior Cloning + RL | 12M ($\mathcal{E}_{\text{train}}$) | Dense | $\hat{\mathcal{V}}_{\text{yt}}$ | $0.59 \pm 0.06$ | $0.79 \pm 0.04$ | $0.66 \pm 0.06$ | $0.67 \pm 0.07$ | $0.30 \pm 0.04$ |
| Our (Value Learning from Videos) | 40K ($\mathcal{E}_{\text{train}}$) | - | $\hat{\mathcal{V}}_{\text{yt}}$ | $\mathbf{0.76} \pm 0.05$ | $\mathbf{0.94} \pm 0.03$ | $\mathbf{0.86} \pm 0.04$ | $\mathbf{0.91} \pm 0.04$ | $0.57 \pm 0.05$ |
| Behavior Cloning | 40K ($\mathcal{E}_{\text{train}}$) | - | $\hat{\mathcal{V}}_{\text{syn}}$ | $0.70 \pm 0.05$ | $0.84 \pm 0.04$ | $0.70 \pm 0.05$ | $0.82 \pm 0.06$ | $0.51 \pm 0.05$ |
| Behavior Cloning + RL | 12M ($\mathcal{E}_{\text{train}}$) | Dense | $\hat{\mathcal{V}}_{\text{syn}}$ | $0.59 \pm 0.06$ | $0.77 \pm 0.04$ | $0.60 \pm 0.06$ | $0.68 \pm 0.07$ | $0.25 \pm 0.04$ |
| Our (Value Learning from Videos) | 40K ($\mathcal{E}_{\text{train}}$) | - | $\hat{\mathcal{V}}_{\text{syn}}$ | $0.70 \pm 0.05$ | $0.90 \pm 0.03$ | $0.77 \pm 0.05$ | $0.82 \pm 0.06$ | $\mathbf{0.62} \pm 0.05$ |
| Strong Supervision Values | Labeled Maps ($\mathcal{E}_{\text{video}}$) | | | $0.72 \pm 0.05$ | $0.91 \pm 0.03$ | $0.86 \pm 0.04$ | $0.94 \pm 0.03$ | $0.71 \pm 0.04$ |
| Strong Supervision + VLV (Ours) | Labeled Maps ($\mathcal{E}_{\text{video}}$) + $\hat{\mathcal{V}}_{\text{yt}}$ | | | $0.81 \pm 0.05$ | $0.94 \pm 0.02$ | $0.82 \pm 0.04$ | $0.94 \pm 0.03$ | $0.69 \pm 0.04$ |

**Table S4: Results**: SPL and Success Rate for ObjectGoal wth **Policy Stop** in novel environments $\mathcal{E}_{\text{test}}$ by episode difficulty. Details in Section 4.2.

| Method | Training Supervision | | | SPL | | | |
|---|---|---|---|---|---|---|---|
| | **# Active Frames** | **Reward** | **Other** | **Easy** | **Medium** | **Hard** | **Overall** |
| Topological Exploration | - | - | - | $0.22 \pm 0.03$ | $0.12 \pm 0.02$ | $0.06 \pm 0.01$ | $0.13 \pm 0.01$ |
| Detection Seeker | - | - | - | $0.31 \pm 0.04$ | $0.22 \pm 0.03$ | $0.09 \pm 0.01$ | $0.19 \pm 0.02$ |
| RL (RGB ResNet) | 20M ($\mathcal{E}_{\text{train}}$) | Dense | - | $0.10 \pm 0.02$ | $0.09 \pm 0.02$ | $0.05 \pm 0.01$ | $0.08 \pm 0.01$ |
| Behavior Cloning | 40K ($\mathcal{E}_{\text{train}}$) | - | $\hat{\mathcal{V}}_{\text{yt}}$ | $0.16 \pm 0.03$ | $0.10 \pm 0.02$ | $0.02 \pm 0.01$ | $0.08 \pm 0.01$ |
| Our (Value Learning from Videos) | 40K ($\mathcal{E}_{\text{train}}$) | - | $\hat{\mathcal{V}}_{\text{yt}}$ | $\mathbf{0.32} \pm 0.04$ | $\mathbf{0.29} \pm 0.03$ | $0.11 \pm 0.02$ | $\mathbf{0.22} \pm 0.02$ |
| Behavior Cloning | 40K ($\mathcal{E}_{\text{train}}$) | - | $\hat{\mathcal{V}}_{\text{syn}}$ | $0.13 \pm 0.02$ | $0.12 \pm 0.02$ | $0.07 \pm 0.01$ | $0.10 \pm 0.01$ |
| Our (Value Learning from Videos) | 40K ($\mathcal{E}_{\text{train}}$) | - | $\hat{\mathcal{V}}_{\text{syn}}$ | $0.29 \pm 0.04$ | $0.23 \pm 0.03$ | $\mathbf{0.13} \pm 0.02$ | $0.21 \pm 0.02$ |
| Strong Supervision Values | Labeled Maps ($\mathcal{E}_{\text{video}}$) | | | $0.34 \pm 0.04$ | $0.25 \pm 0.03$ | $0.15 \pm 0.02$ | $0.24 \pm 0.02$ |
| Strong Supervision + VLV (Ours) | Labeled Maps ($\mathcal{E}_{\text{video}}$) + $\hat{\mathcal{V}}_{\text{yt}}$ | | | $0.31 \pm 0.04$ | $0.29 \pm 0.03$ | $0.13 \pm 0.02$ | $0.23 \pm 0.02$ |

| Method | Training Supervision | | | Success Rate | | | |
|---|---|---|---|---|---|---|---|
| | **# Active Frames** | **Reward** | **Other** | **Easy** | **Medium** | **Hard** | **Overall** |
| Topological Exploration | - | - | - | $0.43 \pm 0.04$ | $0.31 \pm 0.04$ | $0.17 \pm 0.03$ | $0.29 \pm 0.02$ |
| Detection Seeker | - | - | - | $0.52 \pm 0.05$ | $0.43 \pm 0.05$ | $0.21 \pm 0.03$ | $0.37 \pm 0.02$ |
| RL (RGB ResNet) | 20M ($\mathcal{E}_{\text{train}}$) | Dense | - | $0.27 \pm 0.04$ | $0.26 \pm 0.04$ | $0.12 \pm 0.03$ | $0.21 \pm 0.02$ |
| Behavior Cloning | 40K ($\mathcal{E}_{\text{train}}$) | - | $\hat{\mathcal{V}}_{\text{yt}}$ | $0.36 \pm 0.04$ | $0.25 \pm 0.04$ | $0.05 \pm 0.02$ | $0.20 \pm 0.02$ |
| Our (Value Learning from Videos) | 40K ($\mathcal{E}_{\text{train}}$) | - | $\hat{\mathcal{V}}_{\text{yt}}$ | $\mathbf{0.53} \pm 0.04$ | $\mathbf{0.48} \pm 0.05$ | $0.22 \pm 0.03$ | $\mathbf{0.39} \pm 0.02$ |
| Behavior Cloning | 40K ($\mathcal{E}_{\text{train}}$) | - | $\hat{\mathcal{V}}_{\text{syn}}$ | $0.35 \pm 0.04$ | $0.28 \pm 0.04$ | $0.18 \pm 0.03$ | $0.26 \pm 0.02$ |
| Our (Value Learning from Videos) | 40K ($\mathcal{E}_{\text{train}}$) | - | $\hat{\mathcal{V}}_{\text{syn}}$ | $0.50 \pm 0.05$ | $0.42 \pm 0.05$ | $\mathbf{0.26} \pm 0.03$ | $0.38 \pm 0.02$ |
| Strong Supervision Values | Labeled Maps ($\mathcal{E}_{\text{video}}$) | | | $0.58 \pm 0.05$ | $0.45 \pm 0.05$ | $0.30 \pm 0.04$ | $0.43 \pm 0.02$ |
| Strong Supervision + VLV (Ours) | Labeled Maps ($\mathcal{E}_{\text{video}}$) + $\hat{\mathcal{V}}_{\text{yt}}$ | | | $0.50 \pm 0.05$ | $0.50 \pm 0.05$ | $0.27 \pm 0.04$ | $0.41 \pm 0.02$ |

**Table S5: Results**: SPL and Success Rate for ObjectGoal wth **Policy Stop** in novel environments $\mathcal{E}_{\text{test}}$ by object class. Details in Section 4.2.

| Method | Training Supervision | | | SPL | | | | |
|---|---|---|---|---|---|---|---|---|
| | # Active Frames | Reward | Other | Bed | Chair | Couch | Dining Table | Toilet |
| Topological Exploration | - | - | - | $0.18 \pm 0.04$ | $0.17 \pm 0.03$ | $0.09 \pm 0.02$ | $0.11 \pm 0.03$ | $0.09 \pm 0.02$ |
| Detection Seeker | - | - | - | $\mathbf{0.25} \pm 0.04$ | $0.25 \pm 0.04$ | $0.13 \pm 0.03$ | $\mathbf{0.23} \pm 0.05$ | $0.14 \pm 0.02$ |
| RL (RGB ResNet) | 20M ($\mathcal{E}_{\text{train}}$) | Dense | - | $0.02 \pm 0.01$ | $0.15 \pm 0.03$ | $0.05 \pm 0.02$ | $0.07 \pm 0.03$ | $0.08 \pm 0.02$ |
| Behavior Cloning | 40K ($\mathcal{E}_{\text{train}}$) | - | $\hat{\mathcal{V}}_{\text{yt}}$ | $0.13 \pm 0.03$ | $0.12 \pm 0.03$ | $0.02 \pm 0.01$ | $0.12 \pm 0.04$ | $0.05 \pm 0.02$ |
| Our (Value Learning from Videos) | 40K ($\mathcal{E}_{\text{train}}$) | - | $\hat{\mathcal{V}}_{\text{yt}}$ | $\mathbf{0.25} \pm 0.04$ | $\mathbf{0.28} \pm 0.04$ | $\mathbf{0.22} \pm 0.04$ | $0.20 \pm 0.05$ | $0.17 \pm 0.03$ |
| Behavior Cloning | 40K ($\mathcal{E}_{\text{train}}$) | - | $\hat{\mathcal{V}}_{\text{syn}}$ | $0.06 \pm 0.02$ | $0.16 \pm 0.03$ | $0.09 \pm 0.02$ | $0.14 \pm 0.04$ | $0.08 \pm 0.02$ |
| Our (Value Learning from Videos) | 40K ($\mathcal{E}_{\text{train}}$) | - | $\hat{\mathcal{V}}_{\text{syn}}$ | $0.21 \pm 0.04$ | $0.25 \pm 0.03$ | $0.13 \pm 0.03$ | $0.17 \pm 0.04$ | $\mathbf{0.24} \pm 0.03$ |
| Strong Supervision Values | Labeled Maps ($\mathcal{E}_{\text{video}}$) | | | $0.24 \pm 0.04$ | $0.24 \pm 0.03$ | $0.22 \pm 0.04$ | $0.28 \pm 0.05$ | $0.22 \pm 0.03$ |
| Strong Supervision + VLV (Ours) | Labeled Maps ($\mathcal{E}_{\text{video}}$) + $\hat{\mathcal{V}}_{\text{yt}}$ | | | $0.14 \pm 0.04$ | $0.31 \pm 0.04$ | $0.18 \pm 0.04$ | $0.32 \pm 0.05$ | $0.24 \pm 0.03$ |

| Method | Training Supervision | | | Success Rate | | | | |
|---|---|---|---|---|---|---|---|---|
| | # Active Frames | Reward | Other | Bed | Chair | Couch | Dining Table | Toilet |
| Topological Exploration | - | - | - | $0.27 \pm 0.05$ | $0.35 \pm 0.05$ | $0.26 \pm 0.05$ | $0.29 \pm 0.07$ | $0.27 \pm 0.04$ |
| Detection Seeker | - | - | - | $0.38 \pm 0.06$ | $0.48 \pm 0.05$ | $0.23 \pm 0.05$ | $\mathbf{0.37} \pm 0.07$ | $0.36 \pm 0.05$ |
| RL (RGB ResNet) | 20M ($\mathcal{E}_{\text{train}}$) | Dense | - | $0.07 \pm 0.03$ | $0.38 \pm 0.05$ | $0.12 \pm 0.04$ | $0.15 \pm 0.05$ | $0.23 \pm 0.04$ |
| Behavior Cloning | 40K ($\mathcal{E}_{\text{train}}$) | - | $\hat{\mathcal{V}}_{\text{yt}}$ | $0.23 \pm 0.05$ | $0.30 \pm 0.05$ | $0.06 \pm 0.03$ | $0.33 \pm 0.07$ | $0.14 \pm 0.03$ |
| Our (Value Learning from Videos) | 40K ($\mathcal{E}_{\text{train}}$) | - | $\hat{\mathcal{V}}_{\text{yt}}$ | $\mathbf{0.40} \pm 0.06$ | $\mathbf{0.52} \pm 0.05$ | $\mathbf{0.36} \pm 0.06$ | $0.31 \pm 0.07$ | $0.32 \pm 0.04$ |
| Behavior Cloning | 40K ($\mathcal{E}_{\text{train}}$) | - | $\hat{\mathcal{V}}_{\text{syn}}$ | $0.11 \pm 0.04$ | $0.43 \pm 0.05$ | $0.24 \pm 0.05$ | $0.34 \pm 0.07$ | $0.20 \pm 0.04$ |
| Our (Value Learning from Videos) | 40K ($\mathcal{E}_{\text{train}}$) | - | $\hat{\mathcal{V}}_{\text{syn}}$ | $0.30 \pm 0.05$ | $0.49 \pm 0.05$ | $0.26 \pm 0.05$ | $0.34 \pm 0.07$ | $\mathbf{0.43} \pm 0.05$ |
| Strong Supervision Values | Labeled Maps ($\mathcal{E}_{\text{video}}$) | | | $0.34 \pm 0.06$ | $0.50 \pm 0.05$ | $0.37 \pm 0.05$ | $0.54 \pm 0.07$ | $0.42 \pm 0.05$ |
| Strong Supervision + VLV (Ours) | Labeled Maps ($\mathcal{E}_{\text{video}}$) + $\hat{\mathcal{V}}_{\text{yt}}$ | | | $0.20 \pm 0.05$ | $0.53 \pm 0.05$ | $0.30 \pm 0.05$ | $0.52 \pm 0.08$ | $0.47 \pm 0.05$ |

## S3.2 Ablations (corresponding to Section 4.2)

**Table S6:** We report various ablations of our method, when using automatic stopping behavior, evaluated on $\mathcal{E}_{\text{train}}$. Base setting uses noisy trajectores, action labels from inverse models and panorama images. We ablate these settings. See Section 4.3 for details.

| | SPL | | | | Success Rate | | | |
| Method | Easy | Medium | Hard | Overall | Easy | Medium | Hard | Overall |
|---|---|---|---|---|---|---|---|---|
| Base setting | 0.62± 0.04 | 0.42± 0.04 | 0.23± 0.03 | 0.40± 0.02 | 0.95± 0.03 | 0.86± 0.05 | 0.56± 0.05 | 0.75± 0.03 |
| True actions | 0.61± 0.05 | 0.45± 0.05 | 0.25± 0.03 | 0.41± 0.03 | 0.94± 0.03 | 0.86± 0.05 | 0.51± 0.05 | 0.73± 0.03 |
| True detections | 0.62± 0.05 | 0.45± 0.05 | 0.22± 0.03 | 0.40± 0.03 | 0.95± 0.03 | 0.86± 0.05 | 0.48± 0.05 | 0.72± 0.03 |
| True rewards | 0.64± 0.05 | 0.46± 0.05 | 0.21± 0.03 | 0.41± 0.03 | 0.95± 0.03 | 0.86± 0.05 | 0.48± 0.05 | 0.72± 0.03 |
| No noise in videos | 0.65± 0.05 | 0.46± 0.04 | 0.25± 0.03 | 0.43± 0.03 | 0.95± 0.03 | 0.92± 0.04 | 0.59± 0.05 | 0.78± 0.03 |
| $D_{\text{coco}}$ score | 0.73± 0.04 | 0.48± 0.05 | 0.26± 0.03 | 0.46± 0.03 | 0.98± 0.02 | 0.88± 0.05 | 0.58± 0.06 | 0.78± 0.03 |
| Train on $360°$ videos | 0.66± 0.04 | 0.51± 0.05 | 0.32± 0.03 | 0.47± 0.02 | 0.98± 0.02 | 0.92± 0.04 | 0.66± 0.05 | 0.82± 0.03 |

# S4 Visualizations

## S4.1 Value Predictions on Panorama

| | | | | | | | | | | | | |
|---|---|---|---|---|---|---|---|---|---|---|---|---|
| Bed | 0.83 | 0.85 | 0.85 | 0.86 | 0.86 | 0.83 | 0.80 | 0.79 | 0.85 | 0.84 | 0.84 | 0.83 |
| Chair | 0.83 | 0.84 | 0.83 | 0.85 | 0.87 | 0.93 | 0.93 | 0.93 | 0.90 | 0.86 | 0.85 | 0.84 |
| Couch | 0.73 | 0.73 | 0.73 | 0.74 | 0.75 | 0.85 | 0.90 | 0.94 | 0.84 | 0.74 | 0.74 | 0.72 |
| D. Table | 0.70 | 0.71 | 0.71 | 0.72 | 0.75 | 0.80 | 0.82 | 0.82 | 0.80 | 0.75 | 0.72 | 0.70 |
| Toilet | 0.84 | 0.83 | 0.84 | 0.75 | 0.70 | 0.69 | 0.67 | 0.64 | 0.73 | 0.85 | 0.85 | 0.84 |

| | | | | | | | | | | | | |
|---|---|---|---|---|---|---|---|---|---|---|---|---|
| Bed | 0.78 | 0.78 | 0.81 | 0.83 | 0.87 | 0.91 | 0.93 | 0.84 | 0.78 | 0.80 | 0.79 | 0.80 |
| Chair | 0.97 | 0.96 | 0.98 | 0.91 | 0.85 | 0.85 | 0.89 | 0.93 | 0.99 | 0.97 | 0.99 | 0.99 |
| Couch | 1.00 | 1.00 | 1.01 | 0.89 | 0.77 | 0.74 | 0.82 | 0.81 | 0.91 | 0.94 | 0.94 | 0.98 |
| D. Table | 0.81 | 0.80 | 0.79 | 0.78 | 0.72 | 0.71 | 0.77 | 0.83 | 0.87 | 0.89 | 0.85 | 0.81 |
| Toilet | 0.59 | 0.60 | 0.62 | 0.63 | 0.64 | 0.64 | 0.64 | 0.64 | 0.59 | 0.60 | 0.59 | 0.63 |

| | | | | | | | | | | | | |
|---|---|---|---|---|---|---|---|---|---|---|---|---|
| Bed | 0.84 | 0.84 | 0.85 | 0.87 | 0.87 | 0.87 | 0.88 | 0.91 | 0.90 | 0.87 | 0.87 | 0.86 |
| Chair | 0.85 | 0.89 | 0.90 | 0.89 | 0.88 | 0.91 | 0.92 | 0.90 | 0.91 | 0.87 | 0.88 | 0.85 |
| Couch | 0.78 | 0.79 | 0.92 | 0.92 | 0.82 | 0.86 | 0.77 | 0.76 | 0.78 | 0.77 | 0.79 | 0.76 |
| D. Table | 0.72 | 0.73 | 0.78 | 0.76 | 0.74 | 0.76 | 0.79 | 0.75 | 0.77 | 0.73 | 0.76 | 0.72 |
| Toilet | 0.98 | 0.95 | 0.82 | 0.75 | 0.79 | 0.81 | 0.82 | 0.81 | 0.81 | 0.83 | 0.84 | 0.93 |

**Figure S3:** Example panoramas from novel environments with scores from our value network. Scores for each object class (Bed, Chair, Couch, Dining Table, and Toilet) are reported. We can see that value is high in the likely direction of objects even if the object is not directly visible.

| | | | | | | | | | | | | |
|---|---|---|---|---|---|---|---|---|---|---|---|---|
| Bed | 0.83 | 0.82 | 0.80 | 0.81 | 0.80 | 0.79 | 0.78 | 0.82 | 0.81 | 0.84 | 0.85 | 0.84 |
| Chair | 0.96 | 0.98 | 0.98 | 0.98 | 0.98 | 0.98 | 0.99 | 0.98 | 0.97 | 0.96 | 0.93 | 0.93 |
| Couch | 0.79 | 0.80 | 0.80 | 0.79 | 0.85 | 0.86 | 0.91 | 0.93 | 0.92 | 0.89 | 0.79 | 0.80 |
| D. Table | 0.90 | 0.97 | 0.94 | 0.93 | 0.90 | 0.96 | 0.88 | 0.83 | 0.85 | 0.88 | 0.82 | 0.83 |
| Toilet | 0.70 | 0.67 | 0.64 | 0.66 | 0.65 | 0.63 | 0.64 | 0.66 | 0.71 | 0.72 | 0.73 | 0.71 |

| | | | | | | | | | | | | |
|---|---|---|---|---|---|---|---|---|---|---|---|---|
| Bed | 0.78 | 0.78 | 0.80 | 0.80 | 0.83 | 0.83 | 0.84 | 0.84 | 0.84 | 0.83 | 0.80 | 0.78 |
| Chair | 0.99 | 0.99 | 0.99 | 1.00 | 0.99 | 0.96 | 0.92 | 0.96 | 0.97 | 0.98 | 0.97 | 0.99 |
| Couch | 0.99 | 0.95 | 0.84 | 0.80 | 0.82 | 0.82 | 0.80 | 0.84 | 0.87 | 0.90 | 0.94 | 0.99 |
| D. Table | 0.87 | 0.97 | 0.99 | 1.01 | 0.92 | 0.88 | 0.82 | 0.84 | 0.85 | 0.85 | 0.84 | 0.83 |
| Toilet | 0.62 | 0.63 | 0.65 | 0.63 | 0.71 | 0.68 | 0.71 | 0.71 | 0.71 | 0.66 | 0.63 | 0.62 |

| | | | | | | | | | | | | |
|---|---|---|---|---|---|---|---|---|---|---|---|---|
| Bed | 0.84 | 0.81 | 0.80 | 0.82 | 0.83 | 0.82 | 0.84 | 0.83 | 0.84 | 0.83 | 0.84 | 0.84 |
| Chair | 0.96 | 0.94 | 0.95 | 0.96 | 0.95 | 0.92 | 0.92 | 0.86 | 0.88 | 0.87 | 0.88 | 0.95 |
| Couch | 0.84 | 0.81 | 0.80 | 0.84 | 0.86 | 0.78 | 0.76 | 0.75 | 0.75 | 0.76 | 0.76 | 0.83 |
| D. Table | 0.88 | 0.90 | 0.89 | 0.89 | 0.84 | 0.81 | 0.80 | 0.75 | 0.75 | 0.76 | 0.76 | 0.87 |
| Toilet | 0.71 | 0.67 | 0.66 | 0.66 | 0.69 | 0.76 | 0.80 | 0.80 | 0.83 | 0.80 | 0.79 | 0.75 |

**Figure S4:** Example panoramas from novel environments with scores from our value network. Scores for each object class (Bed, Chair, Couch, Dining Table, and Toilet) are reported. We can see that value is high in the likely direction of objects even if the object is not directly visible.

## S4.2   Executed Trajectories

**Figure S5:** Example trajectores from our method navigating in novel environments, sorted by SPL (first few show successes, last few show failures). The black path indicates the trajectory taken by the agent. A blue circle indicates potential short-term goal, and a red rectangle indicates the object goal.

## S4.3  $\mathcal{E}_{\text{test}}$ Problem Setup Visualization

**Figure S6:** Top-down maps of selected floors from the $\mathcal{E}_{\text{test}}$ environments. We also show ground truth object locations. Agent does not have access to any of these maps or ground truth object locations. Visualizations here are provided only to show the difficulty and realism of our problem setup.

## S4.4  Learned Value Maps on Held-out Environments

**Figure S7:** Maps representing the value of different locations in novel environments as predicted by our method trained on $\hat{\mathcal{V}}_{\text{syn}}$. We can see that high value regions fall off smoothly as the distance from object goals increases.

## S4.5  Value in Branching Environment

**Figure S8:** The predicted value in the branching environment using models trained with Q-learning, and policy evaluation via TD(0) and Monte Carlo. We see that the policy evaluation methods drastically under estimate the value in the optimal direction at the branch point. This leads to sub-optimal policies for those methods while the Q-learning based value function finds the optimal trajectory. See Section 4.3 for details.

## Footnotes

[1] As we keep popping values from the priority heap, there are $11N + 1$ (and not $12N$) entries in the heap at the popping time.