[Reviews · NeurIPS 2020]

Review 1

Summary and Contributions: The paper proposes an approach to learning improved semantic navigation policies in indoor environments by building a pseudo-labeled dataset from YouTube videos. The approach requires less labeled data than previous work and makes use of a small action prediction model to learn a strong high-level planning policy, then delegating short-term navigation to a standard algorithm.

Strengths: The method requires very few interaction samples compared to other similar approaches and cleverly leverages a large collection of YouTube video. The high-level navigation policy seems to capture the biases common to locations of target objects in scenes and assigns somewhat reasonable values to views it perceives. The inverse model seems effective enough in providing useful data for off-policy Q learning. The empirical evaluation is well-conducted and convincing, demonstrating the effectiveness of the approach.

Weaknesses: It would be nice to see something like this work without the traditional algorithm for shorter-term navigation which requires a lot of sensor information, but I understand that is somewhat outside the scope of the paper's goal (which is more explicitly to learn the higher-level navigation). I would also be interested in seeing whether the policy it learns indoors transfers to outdoors scenes (is this present in the Gibson videos used for evaluation?). Unless I am misunderstanding, I believe this model cannot transfer to new goal objects at test time (i.e. a coffee machine that also has a detector), since it learns a class-specific value network at training. Is it possible to run the algorithm with a fixed-size priority heap, i.e. without keeping a history of the 12 views gathered from each node explored? This seems like a negative of computational and storage cost.

Correctness: Yes, the claims made are reasonable and accurately reflect the nature of the paper.

Clarity: Overall, the paper structure is clear, with sufficient details provided in Supplementary Material to understand the method in detail and its results. Details in the main paper are a bit light at times, but they are provided in Supplementary.

Relation to Prior Work: The discussion of prior work is reasonable. The relationship to [39], which I believe to be the nearest neighbor of this paper, is fairly discussed, and the application of the inverse model to a collection of real-world videos to yield noisy learning signal for navigation is novel to my knowledge.

Reproducibility: Yes

Additional Feedback:


Review 2

Summary and Contributions: Paper presents a novel methodology for learning in-door navigation policies from unlabelled video data. The agent learns to solve the ObjectGoal task (navigating a house to find an object such as a bed) by learning the Q-function from observing YouTube videos of house demostrations. These videos captures semantic about house layouts that provides guidance when navigating a previously unvisited house. The paper demonstrates that their method outperforms other methods such as direct RL and imitation learning. The great drawback is the lack of comparison to other state of the art navigation agents, yet I think the present methodology is interesting enough and orthogonal to previous work that the paper should be accepted.

Strengths: Paper presents a novel and interesting methodology for unsupervised learning of navigation policies from unlabelled video data. It would be valuable for the AI community to able to efficiently leverage the vast amount of unlabelled video data for training embodied agents. This could open up new research avenues. The paper demonstrate that the proposed method shows great promise and outperforms a plethora of baselines, such as RL and behaviour cloning. Paper presents credible ablation studies and analysis of their method that justifies their design decisions.

Weaknesses: Greatest weakness is the lack of comparison to other navigation agents such as, "Learning to Explore using Active Neural SLAM" or similar works in ObjectGoal navigation. While these methods might not be able to utilize the V_yt dataset they can still train in the E_train environments. Without comparing to state-of-the-art methods it is difficult to compare how much is gained by learning from video data in addition to interactive with the environment.

Correctness: As far as I can tell no claims are incorrect and the method is valid. Paper uses standard metrics such as SPL and success rate, as is the norm for the ObjectGoal task.

Clarity: Paper is very well-written, with good illustration, clear examples and good structure. A concluding section is missing which makes the end somewhat abrupt though I suspect this could be added to a camera ready.

Relation to Prior Work: Previous work is discussed in length in the related work section. Both in respect to learning from video as well as visual navigation and hierarchical policies.

Reproducibility: Yes

Additional Feedback: There are typos in "Is action pseudo-labeling necessary?" with G_far being called G_2 and G_near. This section needs to be refined and corrected.


Review 3

Summary and Contributions: This paper proposes to learn semantic cues for navigation from YouTube videos. Towards that goal, q-learning with pseudo-labeled transition quadruples is used. The proposed approach is called Value Learning from Videos (VLV). Experimental studies is conduction in simulation. Followed by [39] a set of 40K sample interactions is used to train an inverse model. Also, video frames are used to train a classifier to check whether it contains the desired target object or not. Finally, a hierarchical navigation approach is used for learning the policy.

Strengths: + Good visualizations are provided + The important problem of robot navigation is considered + Details about experimental setups are provided in the supp. material. + The paper is readable and the writing of the paper is of good quality.

Weaknesses: -The proposed method is not particularly novel compared to the prior work, specially [39]. In particular the underlying hierarchical navigation method proposed in this paper that creates a topological map is very similar to [39], [49] and [6] . -The central contribution of this paper (as also noted in Line 119) is a value function that learns to produce the value of an image observation for reaching an object category. This is not a significant contribution. In fact, a similar value function has previously been proposed in the semantic object navigation work of [a]. Adding citation to the related work of [1.r] is also strongly suggested. [1.r]. Sadeghi, F. "Divis: Domain invariant visual servoing for collision-free goal reaching." RSS 2019. -Experimental evaluations are weak and does not show significant superiority of the proposed approach compared to the baselines. -This work is far from being applicable in the real environments. There is a huge bias towards real-state type videos.

Correctness: -While the proposed method is introduced as a reinforcement learning approach from videos, it heavily relies on annotation data. In addition to that, the YouTube videos are collected by experts. Therefore, it is more correct to pose the proposed approach as an imitation learning solution rather than RL. -The proposed approach is introduced as learning from YouTube videos. However, in the experimental section, it is mentioned that the train set of the Gibson dataset is also used for training the policies in addition to youtube video. Based on this, I think, introducing the novelty of the paper in the abstract and introduction as using youtube videos for learning is not completely accurate. -In terms of experimental results, the performance of the method is marginally better than the basic behavior cloning (BC) also more elaborate version of BC is not tested so it is not clear if the proposed approach has significant benefit over behavior cloning. Also, the simple naive ‘Detection Seeker’ baseline performs very competitively compared with the proposed method (See Table 1, Table S2, Table S3, and Table S4) which shows that simply applying object detection will result in a success in most of the cases.

Clarity: -The paper is readable and the writing of the paper is of good quality. -I did not find Fig. 1 much expressive for the idea presented in this paper. Also, Fig. 2 is unnecessarily big. I’d suggest that space be used for some of the other results/details presented in the supplementary material.

Relation to Prior Work: -While this work is related to the past work on imitation from observation(IfO), this trend of work is not thoroughly discussed. Also, no prior IfO approach is included as a baseline in the experimental studies. There is a large body of work on IfO which is strongly recommended to be discussed in the related work and be included in the experimental evaluations. To point out a few of those work please check references [b], listed below. Adding citation to these past works is strongly recommended. [2.r]. YuXuan Liu, Abhishek Gupta, Pieter Abbeel, and Sergey Levine. Imitation from observation: Learning to imitate behaviors from raw video via context translation. In 2018 IEEE International Conference on Robotics and Automation (ICRA), pages 1118–1125. IEEE, 2018 [4.r]. Brahma S Pavse, Faraz Torabi, Josiah P Hanna, Garrett Warnell, and Peter Stone. Ridm: Reinforced inverse dynamics modeling for learning from a single observed demonstration. arXiv preprint arXiv:1906.07372, 2019. [5.r]. Sermanet, Pierre, et al. "Time-contrastive networks: Self-supervised learning from video." 2018 IEEE International Conference on Robotics and Automation (ICRA). IEEE, 2018. [6.r]. Sermanet, Pierre, Kelvin Xu, and Sergey Levine. "Unsupervised perceptual rewards for imitation learning." RSS 2017 [7.r]. Dwibedi, D., Tompson, J., Lynch, C., & Sermanet, P. "Learning actionable representations from visual observations." 2018 IEEE/RSJ International Conference on Intelligent Robots and Systems (IROS). IEEE, 2018.

Reproducibility: No

Additional Feedback: The novelty of the proposed approach in this paper is not significantly novel compared to past work. Based on the provided experimental results, the proposed approach performs only marginally better than simple baselines of BC and an object detector policy (Detection Seeker). Additionally, closely related baselines (such as IfO approaches mentioned above) should be added to better identify the strength/weaknesses of the approach. Adding citation to the previous highly related peer-reviewed papers (e.g. [1.r, 2.r, 3.r, 4.r, 5.r, 6.r, 7.r] listed above) is strongly recommended in any version of the manuscript.


Review 4

Summary and Contributions: This paper studies the problem of leveraging pre-existing visual data, in the form of videos, to accelerate training RL agents for indoor navigation problems. There are three claimed contributions of this paper: (1) egocentric video navigation dataset, (2) framework for constructing experience-tuples from videos, and (3) a novel hierarchical policy for navigation.

Strengths: The major strength of this work is providing a framework and proof-of-concept on how to take egocentric video and transfer it into a format that can be used by reinforcement learning algorithms (especially learning value functions). They clearly define the problem and itemize the three major hurdles necessary to implement a solution to this problem (no action labels, no known goals/intents, imperfect demonstrations). They address these problems by combining methods, from outlined previous work, into the aforementioned pipeline. This work is highly relevant to the NeurIPS community as we begin exploring methods of bringing RL agents into the real world. Moreover, it brings a new perspective into ways we can deal with the cost of simulation, by leveraging a new data source (egocentric video feeds).

Weaknesses: The first weakness of this work is the lack of analysis of the overall video-to-experience framework. Each component in this pipeline can introduce error(s) and assumption(s) that must be carefully considered and analyzed. It would greatly aid this work to include discussion of the assumptions taken on by each component, provide discussion about error introduced by each component, and discuss alternative components (and why the chosen ones were used over them). As an example, for the inverse dynamics model: What are the “handful of environments” that are used to train the inverse dynamics model? How different are they from the evaluation setting? It seems like for youtube videos you could construct ground truths trivially by looking at optical flow (or Turk). Did you investigate these methods and/or how does your model compare to the ground truth? What is the impact of errors from the dynamics model on the final overall system? There are also several areas for potential weaknesses in the experimental evaluation: There is a lack of reasonable baseline/ablation methods. There are a lot of different moving parts in the overall “Ours” model, it would be nice to see how the model would perform if each piece was swapped for a ground-truth version, to better understand who the overall system performs. Additionally, statistical baselines could be constructed to better understand how much is learned (e.g., do the houses have a common pattern? chairs/tables are often near each other, should just dining rooms first?). A similar baseline could be BFS/DFS search. I am suspicious about the interpretation of the result metrics based on the two “stopping mechanisms”. How do we know when the agent triggers “Oracle stop” it recognizes that it’s even near the object? For the policy stop, why is the choice to complete not part of the policy? The agent never “chooses” to be done. It’s worth noting for both of these stopping mechanisms the required 1 meter to the object distance seems extremely generous. How does the performance vary when this value is changed? The RL baselines only vary in the design of the neural network architecture. This is a reasonable way to evaluate an end-to-end learned system. However, it would be more fair to show the spectrum of RL models when given access to the spectrum of pretrained neural network modules that are constructed and put in the proposed policy. This would show stronger evidence that the proposed policy offers more advantages than just piggy-backing off of the pretraining. The performance between “Detection Seeker” and the proposed methods are very similar. How should the reader interpret the negligible difference?

Correctness: The claims and methods appear mostly correct in the paper. However, there are most potentially misleading phrasings in the interpretations. In the introduction, the performance boost is only mentioned with respect to a very different model, and no mention is made of the other baselines. Then in the results section similarly should be checked to make sure that it’s not over-claiming the set of methods that each performs better than.

Clarity: The paper is reasonably well written.

Relation to Prior Work: The authors discuss most of the previous work thoroughly. I would strongly encourage the authors additionally place their work in the context of goal-conditioned value functions, or generalized value functions. This will help improve the value function discussion.

Reproducibility: Yes

Additional Feedback: In the broader-impact discussion, please also include discussions about potential issues with using datasets scraped from the internet and the systematic bias present in this process. Of particular note, you mention that previous work majorly focuses on more expensive houses; however, it's reasonable to assume that there's a large underlying bias in the types of houses that have video footage of walkthroughs.

[Author Response · NeurIPS 2020]

**Semantic Visual Navigation by Watching YouTube Videos.** We thank the reviewers for their thoughtful comments.
We are glad that the reviewers share our excitement about the paper, and found it "valuable for the AI community" (**R2**),
"highly relevant to NeurIPS" (**R4**), "brings a new perspective" (**R4**), and one that "could open up new research avenues"
(**R2**). Reviewers also found our methodology "novel and interesting...showing great promise and outperforming a
plethora of baselines" (**R2**), "empirical evaluation well-conducted and convincing" (**R1**), with "credible ablation studies
and analysis" (**R2**). Reviewers also found the paper to be well-written (**R1**, **R2**, **R4**). We address comments from **R1**,
**R2**, and **R4**, and will appropriately incorporate them in final version. Unfortunately, **R3**'s review misrepresents our
work in multiple ways (see below). We point out these factual errors, and urge the **AC** to view **R3**'s review in this light.

**Response to R1.** **New Goal Objects**: Our model can't adapt to new objects at test time, but given a detector for the
new class, we can re-train without extra annotations or environment interactions. **Outdoors**: Videos in $\mathcal{V}_{yt}$ have outdoor
shots (patio, yards, which we currently filter out), thus our models, when retrained, could work well in those outdoor
contexts, but not as well elsewhere. We can't evaluate as Gibson doesn't have outdoor environments. **Use of alternate**
**algorithms for short-term navigation** is possible, as our policy is modular. Though, as noted, it is orthogonal. **Fixed**
**Size Heap?**: Yes! Current episodes only popped $\leq 25$ items. Larger envs / farther goals would need a larger heap.

**Response to R2.** **Comparison to ObjectGoal SOTA**: "Active Neural SLAM [11]" doesn't tackle ObjectGoal (but only
free-space exploration & PointGoal). Our Strong Supervision Value Function (L291) can be thought of as an ObjectGoal
extension of the more recent CVPR 2020 "Neural Topological SLAM [12]". When trained on 85 environments from
$\mathcal{E}_{video}$ with strong supervision, it achieves an OS-SPL of 0.528 (Tab. 1). When combined with $D_{coco}$ this goes up to
0.547. In preliminary experiments done for rebuttal, jointly training this strong supervision value function with our
method on $\mathcal{V}_{yt}$ in a multi-task manner achieves an OS-SPL of 0.563 ($p$-value of 0.098 over the 0.547). Thus, learning
from videos leads to improvements on top of recent methods even when they use strong supervision in 85 environments.

**Response to R4.** **Component design choices, lack of ablations**: We already provide ablations in Sec. 4.2 (and Tab.
S6 in Supp) where, as suggested, we swap in ground-truth versions for each component (in $\mathcal{V}_{syn}$, as $\mathcal{V}_{yt}$ is unlabeled).
Using ground truth versions only lead to a 1%–3% improvement in final performance. This makes us confident in our
design choices. We will augment these existing experiments with the rationale for our design choices. Inverse dynamic
model was trained on *held-out* 15 envs and worked well (acc. 95%), more details on L223, L195, Tab. S1 in supp.
**Pre-training in RL**: Great point. We already tried initializing RL policies with ImageNet trained models (L269). That
said, this still ignores $\mathcal{V}_{yt}$ data, and test-time access to detector $D_{coco}$ used by our method. We present two experiments
that control for these. First, we give the RL models access to $\mathcal{V}_{yt}$, by initializing with the model trained using BC in
L280 (which in-turn used ImageNet initialization). This doesn't help much (OS-SPL of $0.24 \pm 0.02$). Second, we test
our model without $D_{coco}$. This achieves an OS-SPL of 0.44, *vs.* 0.29 for the best of RL models. Thus, our approach
isn't just "piggy-backing off of the pretraining." We will include these additional experiments in the final version.
**Improvement over Detection Seeker**: **1.** Under the tighter *paired student t-test*, SPL (standard metric for ObjectGoal
[4]) of our method ($\mathcal{V}_{yt}$ version) is better than Detection Seeker with $p$-values of 0.0006 and 0.068 in Oracle and Policy
Stop settings respectively. Tab. 1 reported 90% confidence intervals (*i.e.* a looser, unpaired test) so as to report all
methods together. **2.** Improvements over Detection Seeker are more evident in hard episodes (where agents starts far
from target object), as seen in SPL breakdown across episode hardness (Sect. S2.2 & Tab. S2 in supp). It is difficult to
improve upon Detection Seeker in easy cases when the object is likely already in sight from around the start point.
**Other**: Trends are similar at success threshold 0.5m (OS-SPL: Ours 0.34 / Det. Seeker 0.31). Note, in policy stop
setting behavior is *completely* autonomous. Visualizations in Fig. 3,S3,S4,S7 show what our model learned. We will add
generalized value function references, correct the potentially misleading phrasing, & further discuss broader impact.

**Response to R3.** "...central contribution (as noted in L119) is a value function ...": L119 explicitly notes that our
novelty is in *the use of videos* for learning value functions. This differentiates it from [1.r], other prior works [28,42,65].
"...not novel compared to [39]. In particular, ...topological map is very similar to [39, 49, 6].": This is wrong,
[39, 6] do not even use topological maps!! [49] does, but for a different task (going to an image goal in a pre-explored
environment), and without the high-level semantic value function as we do. [39] is related for a different reason. L98
explicitly describes this relationship, which **R1** notes as being "fairly discussed."
"...no prior IfO approach included as a baseline in the experimental studies ...": Our Behavior Cloning on Pseudo
Labeled Videos (L280) is precisely the BCO(0) algorithm from prior IfO work [57]. We achieve relative improvement of
$33\% - 250\%$ SPL over this baseline. We will note this relationship to [57]. References [2.r–7.r] fall in exactly the same
category as IfO references [5,18,57,58] discussed on L105: tackling the same task in the same environment depicted in
the demonstration *vs.* our work that solves novel tasks in novel envs. Furthermore, [2.r,5.r,6.r] obtain policies through
RL on reward functions learned from task demonstrations. Thus, their performance is upper-bounded by that of using
dense ground truth rewards (already in Tab. 1, OS-SPL of 0.29, *vs.* 0.50 for Ours). We will cite [1.r–7.r].
"...marginally better than behavior cloning ...Detection Seeker performs competitively ...": $33\% - 250\%$
relative improvement in SPL over behavior cloning (0.24 *vs.* 0.50, 0.06 *vs.* 0.21, 0.36 *vs.* 0.48, 0.10 *vs.* 0.21, Tab 1.)
isn't marginal!! Detection Seeker performs competitively, but our gains over it are significant (see **L33** above).

[Meta-Review · NeurIPS 2020]

This paper proposes to leverage (mostly real-estate) unlabelled YouTube videos of egocentric navigation in indoor environments, to train the Q value function network for the high-level part of a hierarchical RL policy for goal-driven indoor robot navigation. The lower-level part relies on depth-based obstacle avoidance and planning in 2D maps. The method works in an unsupervised way by relying on two ways of augmenting the egocentric navigation video dataset: 1) extract action labels from motion classifiers and 2) extract semantic goal labels from object detection. It uses these two to 3) build experience replay tuples of (previous image, action, next image, goal) and then train the goal-conditional value function using Q-Learning. The high-level policy predicts Q values for navigating a topological graph. The paper builds upon [39] "Learning navigation subroutines by watching videos" which introduces the first idea (extract action labels from navigation videos) as well as a simpler version of the third idea (collect tuples of previous image, action, next image, in the context of Q-Learning with rewards from intrinsic motivation). An independently published paper, [1.r] "DIViS: Domain Invariant Visual Servoing for Collision-Free Goal Reaching", introduces the second idea (images or image categories, as provided by the same MS-COCO object detector, for goal-driven navigation) in a context similar to [39], to train low-level affordance- and obstacle detection-based policy. The paper could be viewed like a combination of [39], [1.r], and of topological graph navigation papers such as [49] "Semi-parametric topological memory for navigation". Paper [6] "Combining optimal control and learning for visual navigation in novel environments" had a different take for object goal-driven hierarchical navigation by using object detectors for waypoints. The paper is extensively evaluated with ablation experiments, beats behavioural cloning and a few heuristic baselines and demonstrates data efficient learning and performance on the Gibson dataset. The algorithm is well documented in the appendix and code is provided. Reviewers R1, R2 and R4 all gave scores of 7 (and judged the research easily reproducible) and during the internal discussion, they all believed the paper should be accepted. Negative points raised by these 3 reviewers were the use of a depth- and occupancy-map based heuristic as low-level policy, the lack of strong RL baselines (addressed in the rebuttal), some remaining questions on data analysis and ablation studies (also addressed in the rebuttal), and the fact the the method was only marginally better than the non-RL but object-detection based Detection Seeker baseline (more on that later). Reviewer R3 strongly disagreed and gave -- then maintained, during discussion -- a score of 2. Some points of criticism seem less valid: * difficulty in reproducing the work (I disagree, as code and an extensive appendix are provided) * lack of applicability in the real world (even though the method is evaluated on a standard indoor navigation environment) * and the fact that some amount of training on Gibson is still necessary (even though the authors claimed that their method achieves significant data efficiency) The following two points of criticism are to my mind valid: * lack of novelty w.r.t. existing work. The method seems like a combination of existing work and ideas (experience replay from videos, object detection for semantic goals, topological graph navigation). At the same time, one could argue that most research is done this way and novelty is difficult to assess and quantify. * strong performance of a heuristic baseline: Detection Seeker, that does not use the learned Q value function, but still uses object detection and spatial consistency. That point is partially addressed in the rebuttal, but I am worried that the problem is actually much larger and is due to the small scale of the environment itself (apartments), and to the way the 360-degree observations are acquired. To reuse the analogy at the beginning of the paper, one often does not need to get up from their table at a restaurant to have an idea of where the toilets are: simply looking around may be enough. I therefore disagree with R3's score of 2 (strong reject), which is not appropriate for that level of work and analysis, and am counting that as score of 5 instead (good work, extensive analysis, interesting combination of ideas though not as novel as claimed, problems with the environment and with how observations are constructed). Ultimately, I would recommend acceptance for this paper as poster and consider it borderline (it ranks 6/16 in my stack of papers). Additional comment: in the broader impact statement, the authors should acknowledge the nature of the dataset used to train the value function: even though it is publicly available, there are privacy issues, consent issues (as these are real-estate videos, and this particular use case was not envisioned by people who uploaded them) and biases (comparable to the biases in Gibson).